# MALT Powers Up Adversarial Attacks

**Odelia Melamed** *            **Gilad Yehudai**†            **Adi Shamir** ‡

## Abstract

Current adversarial attacks for multi-class classifiers choose the target class for a given input naively, based on the classifier's confidence levels for various target classes. We present a novel adversarial targeting method, *MALT - Mesoscopic Almost Linearity Targeting*, based on medium-scale almost linearity assumptions. Our attack wins over the current state of the art AutoAttack on the standard benchmark datasets CIFAR-100 and ImageNet and for a variety of robust models. In particular, our attack is *five times faster* than AutoAttack, while successfully matching all of AutoAttack's successes and attacking additional samples that were previously out of reach. We then prove formally and demonstrate empirically that our targeting method, although inspired by linear predictors, also applies to standard non-linear models.

## 1 Introduction

Neural networks are widely known to be susceptible to adversarial perturbations (Szegedy et al. [2013]), which are typically imperceptible by humans. Many different papers have shown how to construct such attacks, where adding a small perturbation to the input significantly changes the output of the model (Carlini and Wagner [2017], Papernot et al. [2017], Athalye et al. [2018]). To protect from these attacks, researchers have tried to develop more robust models using several different techniques, such as adversarial training using different attacks (Madry et al. [2017], Papernot et al. [2016], Liu et al. [2023], Wang et al. [2023]).

The current state of the art adversarial attack, known as AutoAttack (Croce and Hein [2020b]), combines several different parameter-free attacks, some targeted and some untargeted. AutoAttack currently leads the RobustBench benchmark (Croce et al. [2020]), which is the standard benchmark for adversarial robustness. Notably, the targeted attacks used in AutoAttack pick the adversarial target classes according to the model's confidence levels and attack the top nine classes, even though CIFAR-100 and ImageNet have many more possible target classes. The reason for attacking only a limited number of classes, rather than all possible classes, is computational, as each such attack has a significant running time.

The reason that adversarial examples exist remains a hot topic of debate, specifically, whether it is due to the highly non-linear landscape of neural networks or rather to their local linearity properties. In Goodfellow et al. [2014], the authors provide several strong arguments for the local linearity hypothesis while presenting FGSM, a one-step gradient attack. Following Bubeck et al. [2021], we consider three distance scales around each fixed data point where linearity properties should be studied separately. On the *macroscopic* scale, neural networks are highly non-linear functions and have very complicated decision boundaries. On the other hand, neural networks with ReLU activation are piecewise linear, and thus at the *microscopic* scale (in which no ReLU input changes signs), they can be seen as completely linear functions. The third is the intermediate *mesoscopic* scale, which characterizes the typical distances between inputs to their nearest adversarial examples.

---

*Weizmann Institute of Science, Israel, `odelia.melamed@weizmann.ac.il`

†Center for Data Science, New York University, `gy2219@nyu.edu`

‡Weizmann Institute of Science, Israel, `adi.shamir@weizmann.ac.il`.

38th Conference on Neural Information Processing Systems (NeurIPS 2024).

At this mesoscopic scale, Bubeck et al. [2021] proved that random networks are *locally almost linear*, hinting that for adversarial examples, we can use prediction techniques that are motivated by the behavior of linear functions.

In this paper, we present a novel adversarial attack, *MALT* (Mesoscopic Almost Linearity Targeting), which is five times faster than the current SOTA attack while exceeding its success rate. It uses the common fast APGD attack (Croce and Hein [2020b]) with a new targeting algorithm. The basic idea of MALT is that instead of sorting the target classes only by the confidence levels of the attacked model, *MALT* normalizes the class's confidence by the norm of the row of the Jacobian corresponding to it. This ordering is motivated by the case of linear classifiers that can be fully analyzed, and makes use of the almost linearity of the model at the mesoscopic scale.

Our attack wins over AutoAttack on both CIFAR-100 and ImageNet datasets for different robust models from the standard RobustBench benchmark. In particular, in all of our experiments, we show that the MALT targeting algorithm eliminates the need to use any attacks other than APGD: for every image that AutoAttack successfully attacks, MALT also succeeds while reaching several additional images on which AutoAttacks fails. A major benefit of MALT is that on SOTA robust models, our attack runs *five times faster* than AutoAttack on the ImageNet attack test dataset.

To further support the mesoscopic almost linearity hypothesis presented in Goodfellow et al. [2014] and Bubeck et al. [2021], we demonstrate both theoretically and empirically that our targeting method, though inspired by linearity considerations, also applies to non-linear models. On the theoretical side, we consider the setting of Melamed et al. [2023], Haldar et al. [2024] where the data resides on a low dimensional manifold, and prove that the network remains "almost linear" at the mesoscopic scale around data points, deducing that the ordering provided by our targeting method is preserved in that scale. Empirically, we demonstrate that models tend to be almost linear close to data points by showing that a linear approximation successfully predicts the model's output when taking random or adversarial steps away from a data point.

## 2   Related Works

**Linearity and adversarial attacks**     The local linearity of non-linear classifier in the context of adversarial example was first hypothesized for explaining the adversarial examples' existence (Goodfellow et al. [2014]), presenting the successful FGSM attack. Later, multi-step attacks with higher success rates have shown that the standard classifiers, indeed, are not entirely linear in the mesoscopic scale (Madry et al. [2017], Carlini and Wagner [2017]). Local linearity was also researched in the robustness context, finding linearity constraints increases the classifier robustness (Uesato et al. [2018], Sarkar and Iyengar [2020], Qin et al. [2019]) and the attack transferability (Papernot et al. [2017], Guo et al. [2020]).

**Targeting methods**     As a non-naive targeting method has yet to be proposed, the FAB attack (Croce and Hein [2020a]) and DeepFool attack (Moosavi-Dezfooli et al. [2016]) are using a step-wise targeting method to boost the untargeted version of the attack. The step-wise point of view, looking at the microscopic scale, allows assuming linearity, and estimating the best target becomes an easy linear problem. Sarkar and Iyengar [2020] uses the same simple method for linearity-enforced classifiers. Our targeting method implements the same linearity-based targeting idea but is calculated once at the starting point.

**Theory of adversarial attacks**     The source of adversarial examples has been extensively researched in recent years. Building on Shamir et al. [2019], it was shown in Daniely and Shacham [2020] that adversarial perturbations appear in random networks. This result was extended to a larger family of networks with random weights (Bartlett et al. [2021], Bubeck et al. [2019, 2021], Montanari and Wu [2022]). In Vardi et al. [2022], Frei et al. [2024], the authors show that the gradient method is implicitly biased towards non-robust networks. The hypothesis of the data lies in a low-dimensional manifold, which the adversarial examples are perpendicular to was suggested in both theoretical (Gilmer et al. [2018], Tanay and Griffin [2016], Melamed et al. [2023]) and applied (Song et al. [2017], Stutz et al. [2019], Shamir et al. [2021]) settings. Melamed et al. [2023] proved that for data that lies on a linear subspace, there exist adversarial perturbations. Their result is extended in Haldar et al. [2024] to clustered data.

# 3    MALT – Mesoscopic Almost Linearity Targeting

The targeted attacks in AutoAttack choose nine target classes to attack, while ImageNet, for example, contains 1000 classes. This is done since the running time of each such targeted attack is long, and attacking each image in the entire ImageNet test dataset for a large number of classes will be cumbersome. Hence, it is extremely important to choose the right nine target classes. Otherwise, the attack may fail on those target classes, while it may succeed on other classes that are not chosen.

The naive targeting in AutoAttack chooses the top nine classes sorted by the output of the model, which corresponds to the confidence the model gives to each class. In Figure 1, we show two examples of images from the ImageNet dataset on which AutoAttack fails, both with targeted and untargeted attacks, while MALT succeeds. For each image $\mathbf{x}$, after finding the adversarial perturbation $\mathbf{v}$ we divide it into 100 equal parts, namely $\mathbf{v}_i = \frac{i}{100}\mathbf{v}$ for $i = 0, \dots, 100$. We plot the network's confidence levels for each class $N(\mathbf{x} + \mathbf{v}_i)$ (y-axis) w.r.t the interval $i = 0, ..., 100$ (x-axis). The advantage of MALT here is that it allows targeting classes that are overlooked by AutoAttack. Before presenting the MALT attack, we motivate it by analyzing adversarial attacks in linear models. Additional examples where MALT finds adversarial examples while AutoAttack fails can be found in Appendix A.

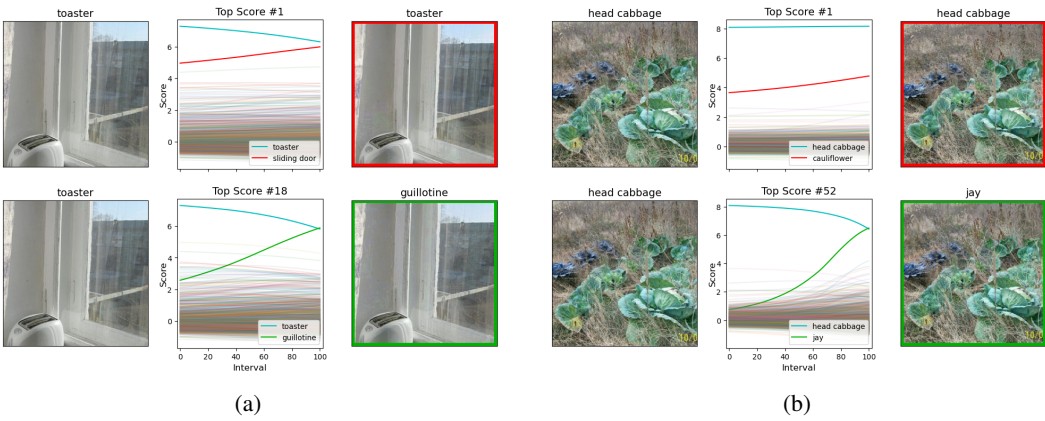

(a)                                                                      (b)

Figure 1: Examples of images from the ImageNet dataset that AutoAttack fails to attack while MALT succeeds. The top row shows an APGD attack on the target class with the highest logit, and the bottom row shows an APGD attack on the class which MALT finds and succeeds, corresponding to the (a) 18th and (b) 52nd classes with the highest logits. The images are shown before and after the attack, and the change in logits is presented in the middle column.

## 3.1    Motivation – The best target in a linear function

Naive targeting methods consider the top-$c$ classes with the largest output of the model, i.e., the top-$c$ logits (often, for $c = 9$), which corresponds to the classes that receive the highest confidence from the model. However, our goal is to find the target class for which its adversarial perturbation is the smallest from a given data point. In fact, it is not the case in general that the targets with the highest confidence will also have a small adversarial perturbation. For linear predictors, it is possible to fully analyze and find the target class with the smallest adversarial perturbation, as shown in the following:

**Lemma 3.1.** *Consider a linear predictor over $k$ classes of the form $F(\mathbf{x}) = W\mathbf{x} + \mathbf{b}$ where $\mathbf{x} \in \mathbb{R}^d, W \in \mathbb{R}^{k \times d}$ and $\mathbf{b} \in \mathbb{R}^k$. Denote the $i$-th row of $W$ by $\mathbf{w}_i$ and by $F_i(\mathbf{x}) = \langle \mathbf{w}_i, \mathbf{x} \rangle + b_i$ the $i$-th output of $F$ for every $i \in [k]$. Let $\mathbf{x}_0 \in \mathbb{R}^d$ with $\arg\max_i F_i(\mathbf{x}_0) = \ell$ and denote by $\epsilon_i := \frac{\langle \mathbf{w}_i - \mathbf{w}_\ell, \mathbf{x}_0 \rangle}{\|\mathbf{w}_i - \mathbf{w}_\ell\|^2}$. Then, the adversarial perturbation $\mathbf{z}$ that changes the label of $\mathbf{x}_0$ with the smallest $L_2$ norm is equal to $\mathbf{z} := \epsilon_j(\mathbf{w}_j - \mathbf{w}_\ell)$ for the target class $j := \arg\min_i \frac{\langle \mathbf{w}_i - \mathbf{w}_\ell, \mathbf{x}_0 \rangle}{\|\mathbf{w}_i - \mathbf{w}_\ell\|}$.*

The full proof is deferred to Appendix B. The above lemma states that, for a data point $\mathbf{x}_0$ classified by the linear predictor as class $\ell$, the class $i$ with the smallest linear perturbation will minimize the term $\frac{\langle \mathbf{w}_i - \mathbf{w}_\ell, \mathbf{x}_0 \rangle}{\|\mathbf{w}_i - \mathbf{w}_\ell\|}$. Note that the term $\langle \mathbf{w}_i - \mathbf{w}_\ell, \mathbf{x}_0 \rangle$ represents the distance between the output of

the model on class $\ell$ and class $i$, which are the logits that are commonly used in selecting the targets for adversarial attacks. The lemma states that this term should be divided by $\|\mathbf{w}_i - \mathbf{w}_\ell\|$, which corresponds to the norm of the difference between the gradients.

There is an intuitive explanation as to why this targeting method is optimal for linear predictors. While the term $\langle \mathbf{w}_i - \mathbf{w}_\ell, \mathbf{x}_0 \rangle$ represents the "distance" that the adversarial perturbation should move to change the prediction class, the term $\|\mathbf{w}_i - \mathbf{w}_\ell\|$ represents the "speed" at which this change happens. Thus, the fastest way to change the prediction class is to consider the target for which the ratio between the distance and the speed is the smallest.

## 3.2 Targeting method

We now formalize MALT, which is an adversarial attack that is applicable to any model, not only linear. The basic idea is to re-order the target classes using our targeting method, and then employ a fast targeted adversarial attack towards those targets.

Consider a classification model over $k$-classes (e.g., a neural network) $N : \mathbb{R}^d \to \mathbb{R}^k$. Suppose we are given a data point $\mathbf{x}_0 \in \mathbb{R}^d$ classified in class $\ell$ by $N$, meaning that $\ell = \arg\max_{j \in \{1,\dots,k\}} (N(\mathbf{x}_0))_j$. We look at the top $c \in \mathbb{N}$ ($c \leq k$) classes ordered by their output on $\mathbf{x}_0$, and for each such class $j$ calculate a score of the form $\frac{(N(\mathbf{x}_0))_j - (N(\mathbf{x}_0))_\ell}{\|(\nabla N(\mathbf{x}_0))_j - (\nabla N(\mathbf{x}_0))_\ell\|}$. We now pick the top $a \in \mathbb{N}$ ($a \leq c \leq k$) classes re-ordered by our score and perform a targeted adversarial attack towards it. The two hyperparameters in our algorithm are $c$, which represents the number of candidates for which we calculate the score, and $a$, which represents the number of classes we perform a targeted attack towards. The full attack algorithm is presented in Algorithm 1.

Note that to calculate a score for some target class, we need to calculate the gradient of the model on $\mathbf{x}_0$ w.r.t this class. For datasets with many possible classes (e.g. ImageNet) we would like to limit these calculations. In all of our experiments, we used $c = 100$ and $a = 9$.

We note that our targeting method is compatible with any targeted attack. We empirically found that APGD (Croce and Hein [2020b]) with the default DLR loss performs well across datasets and models, improving the current state of the art – AutoAttack (Croce and Hein [2020b]). Full empirical results are in Section 5.

---

**Algorithm 1** MALT attack algorithm

---

**Input:** Trained classification model $N : \mathbb{R}^d \to \mathbb{R}^k$, data point $\mathbf{x}_0 \in \mathbb{R}^d$, hyperparameters $a, c \in \mathbb{N}$.

set $\ell = \arg\max_{j \in \{1,\dots,k\}} (N(\mathbf{x}_0))_j$
Set CandidateClasses $= [0, \dots, 0]$, a list of size $c$
**for** top $c$ classes $i \in \{1 \dots, k\} \setminus \{\ell\}$ ordered by $(N(\mathbf{x}_0))_i$ **do**
    Set AttackScore$_i = \frac{(N(\mathbf{x}_0))_i - (N(\mathbf{x}_0))_\ell}{\|(\nabla N(\mathbf{x}_0))_i - (\nabla N(\mathbf{x}_0))_\ell\|}$
    Append $(i, \text{AttackScore}_i)$ to the CandidateClasses list
**end for**
Sort CandidateClasses list by AttackScore
**for** class $i$ in the top $a$ classes of the CandidateClasses list **do**
    Run TargetedAttack$(N, \mathbf{x}_0, i)$
    **if** Adversarial perturbation found **then**
        return the adversarial perturbation
    **end if**
**end for**

---

## 3.3 Complexity analysis

We now calculate the time complexity of MALT with APGD, and compare it with the time complexity of AutoAttack. We calculate the complexity in terms of forward and backward passes (i.e., gradient calculations). We also consider the default hyperparameters of each adversarial attack, e.g., for APGD, we perform an attack with 100 iterations.

**MALT with APGD.** Calculating the targeting order takes $c$ backward passes, since the corresponding row of the Jacobian is calculated. For the top $a$ classes in this ordering, we perform an APGD attack which takes $a \cdot 100$ backward passes and $a \cdot 100$ forward passes.

**AutoAttack.** We calculate the complexity of each attack separately: (1) Untargeted APGD with CE loss takes 100 forward and 100 backward passes; (2) Targeted APGD with DLR loss attack the top 9 targets, which takes 900 forward and 900 backward passes; (3) Targeted FAB also attack the top 9 targets, runs for 100 iterations and takes one backward, two forward passes for each iteration, in addition to three forward passes for each target. In total, it takes 900 backward and 1827 forward passes; and (4) Untargeted Square black-box attack for 5000 steps, which takes 1 forward pass each for a total of 5000 forward passes.

Finally, we sum the total complexity, and use the hyperparameters $c = 100$, $a = 9$ for MALT (which are used in our experimental results). MALT takes 1000 backward and 900 forward passes, while AutoAttack takes 1900 backward and 7827 forward passes. Since each forward pass takes approximately the same time as a backward pass, we conclude that MALT requires 1900 passes, while AutoAttack takes 9727 passes, which is more than *five times* faster. In Section 5, we complement this analysis with experiments, showing that also in practice, MALT is on average five times faster than AutoAttack on the ImageNet test dataset.

## 4 Mesoscopic Almost Linearity in Neural Networks

In the previous section, we defined MALT, which relies on normalizing the output of the model by the magnitude of the gradients. This method was motivated by analyzing a linear predictor and finding the target class with the closest adversarial example. However, neural networks are highly non-linear, and the gradient can potentially change significantly when traversing the trajectory from a data point to an adversarial example. In this section, we claim that neural networks behave as though they are almost linear in the mesoscopic scale, in the sense that the norm of the gradient does not change very much when moving from a data point towards an adversarial example. This means that the target classes which MALT chooses to attack remain good target classes also when moving away from the attacked data point.

### 4.1 Almost Linearity for Data Residing on a Low-Dimensional Subspace

In this subsection, we prove theoretically that 2-layer neural networks are almost linear in the mesoscopic scale and in certain directions under the setting where the high-dimensional data lies on a low-dimensional manifold. This setting was studied in several works such as Fawzi et al. [2018], Khoury and Hadfield-Menell [2018], Shamir et al. [2021], Melamed et al. [2023]. In particular, we consider the setting from Melamed et al. [2023], where the authors analyzed two-layer networks, and the data lies on a low-dimensional linear subspace. All the proofs can be found in Appendix C.

**Model.** Our model is a two-layer fully-connected ReLU network $N : \mathbb{R}^d \to \mathbb{R}$ with input dimension $d$ and hidden dimension $m$: $N(\mathbf{x}, \mathbf{w}_{1:m}) = \sum_{i=1}^m u_i \sigma(\mathbf{w}_i^\top \mathbf{x})$, where $\sigma : \mathbb{R} \to \mathbb{R}$ is a non-linear activation, and $\mathbf{w}_{1:m} = (\mathbf{w}_1, \ldots, \mathbf{w}_m)$. We additionally assume that $\sigma$ is $L$-smooth and there exists $\beta > 0$ such that $\sigma'(z) \geq \beta$ for every $z \in \mathbb{R}$. An example of such an activation is a smoothed version of Leaky ReLU. We initialize the first layer using standard Kaiming initialization He et al. [2015], i.e. $\mathbf{w}_i \sim \mathcal{N}\left(\mathbf{0}, \frac{1}{d}I\right)$, and the output layer as $u_i \sim \mathcal{U}\left(\left\{\pm\frac{1}{\sqrt{m}}\right\}\right)$. Note that in standard Kaiming initialization, each $u_i$ would be initialized normally with a standard deviation of $\frac{1}{\sqrt{m}}$. For ease of analysis, we fix these weights to their standard deviation and only randomize the sign (this was also done in Melamed et al. [2023], Bubeck et al. [2021]).

**Data.** Following Melamed et al. [2023], our main assumption on the data is that it lies on a low dimensional linear subspace. Namely, we consider a binary classification dataset $(\mathbf{x}_1, y_1), \ldots, (\mathbf{x}_m, y_r) \in \mathbb{R}^d \times \{\pm 1\}$. We assume there exists a linear subspace $P$ of dimension $\ell < d$ such that $\mathbf{x}_i \in P$ for every $i$.

**Loss and training.** Given a loss function $L : \mathbb{R} \times \mathbb{R} \to \mathbb{R}$ (e.g. MSE, BCE), we consider the optimization problem: $\min_{\mathbf{w}_{1:m}} \sum_{i=1}^r L(y_i, N(\mathbf{x}_i, \mathbf{w}_{1:m}))$, which is optimized using standard gradient descent. For ease of analysis, we assume only the weights of the first layer are trained (i.e. the $\mathbf{w}_i$'s) while the weights of the second layer (i.e. the $u_i$'s) are fixed at their initial values, this was

also done in Melamed et al. [2023]. Since we consider trained networks in our results, we will write $N(\mathbf{x})$, and omit the $\mathbf{w}_{1:m}$ as an input to the network.

**Mesoscopic local linearity.** Following Bubeck et al. [2021], we say that the network $N$ is *mesoscopic locally linear* at a point $\mathbf{x}_0$ with $\|\mathbf{x}\| \leq \sqrt{d}$ if for every $\mathbf{v}$ with $\|\mathbf{v}\| = o(\sqrt{d})$ we have:

$$\|\nabla N(\mathbf{x}_0) - \nabla N(\mathbf{x}_0 + \mathbf{v})\| = o(\|\nabla N(\mathbf{x}_0)\|) \,. \tag{1}$$

We now show an upper bound on the l.h.s of Eq. (1) and a lower bound on the r.h.s of Eq. (1) when projecting the gradients on the orthogonal subspace on which the data lies on. In the following theorems, $\Pi_{P^\perp}$ means an orthogonal projection on the subspace $P^\perp$.

**Theorem 4.1.** *Suppose that the network $N(\mathbf{x}, \mathbf{w}_{1:m}) = \sum_{i=1}^{m} u_i \sigma(\mathbf{w}_i^\top \mathbf{x})$ is trained on a dataset which lies on a linear subspace $P$ of dimension $\ell$. Then, w.p $> 1 - \delta$ over the initialization for every $\mathbf{v} \in P^\perp$ with $\|\mathbf{v}\| \leq R$ and every $\mathbf{x} \in P$ we have that:*

$$\|\Pi_{P^\perp} (\nabla N(\mathbf{x}) - \nabla N(\mathbf{x} + \mathbf{v}))\| \leq 20 L R \left( \sqrt{\frac{\log\left(\frac{m}{\delta}\right)}{d - \ell}} + \frac{\log\left(\frac{1}{\delta}\right)}{m} \right)$$

**Theorem 4.2.** *Under the same assumptions as in Theorem 4.1 and that $d \geq 2 \log\left(\frac{1}{\delta}\right)$, w.p $> 1 - \delta$ over the initialization we have that:*

$$\|\Pi_{P^\perp} (\nabla_\mathbf{x} N(\mathbf{x}))\| \geq \beta \cdot \sqrt{1 - 2\sqrt{\frac{\log\left(\frac{1}{\delta}\right)}{d}}}$$

Using both theorems, we can prove that the network is mesoscopic almost linear in directions orthogonal to the data subspace:

**Corollary 4.1.** *Suppose that the network $N(\mathbf{x}) = \sum_{i=1}^{m} u_i \sigma(\mathbf{w}_i^\top \mathbf{x})$ is trained on a dataset which lies on a linear subspace $P$ of dimensions $\ell$. Let $\mathbf{v} \in P^\perp$ with $\|\mathbf{v}\| \leq R$ and $\mathbf{x} \in P$. Assume that $\beta = \Omega(1)$, $d - \ell = \Omega(d)$, $R = o(\sqrt{d - \ell})$ and $m = \Omega(d - \ell)$, then we have that:*

$$\|\Pi_{P^\perp}(\nabla N(\mathbf{x}_0) - \nabla N(\mathbf{x}_0 + \mathbf{v}))\| = o(\|\Pi_{P^\perp}(\nabla N(\mathbf{x}_0))\|) \,.$$

The reason that the directions orthogonal to the data subspace are interesting, is because it is proven in previous works that in those directions, there exist adversarial perturbations, see for example Melamed et al. [2023], Haldar et al. [2024]. This means that if we consider an adversarial perturbation in those directions, the ordering of the target classes from our targeting method should remain approximately the same throughout the trajectory of the perturbation. This is because the norm of the difference between the gradients is not too big, at least compared to the norm of the gradient itself.

**Remark 4.1** (Assumption in the theoretical part). *For the formal statements to work, we make several simplifying assumptions. In Appendix C.4, we provide further explanation on the necessity of those assumptions and whether they could be relaxed. In a nutshell, most assumption (especially the ones on the activation) are made since we consider a very general setting, where our only assumption on the data is that it lies on a low dimensional manifold, note that we don't assume a more intricate structure or limit the number of training points. By introducing more assumptions on the data, we could relax our other assumptions.*

### 4.2 Empirical local linearity

In this section, we study mesoscopic almost linearity empirically using state of the art robust networks from RobustBench (Croce et al. [2020]), and for both the CIFAR100 and ImageNet datasets. In the first experiment, for each image $\mathbf{x}_0$ in the test dataset, we consider an $\epsilon$-norm ball in $L_\infty$ around it (with $\epsilon = 8/255$ and $4/255$ for CIFAR100 and ImageNet respectively). Our goal is to study how the gradient changes when moving from $\mathbf{x}_0$ to a $\mathbf{x}_0 + \mathbf{v}$, where $\mathbf{v}$ is some direction inside the $\epsilon$-ball.

We use two strategies to choose the direction $\mathbf{v}$. The first is drawing a random direction for each image, and the second is the direction of the gradient at each image, which corresponds to an adversarial direction. After choosing $\mathbf{v}$ we divide it into 100 equal parts, namely $\mathbf{v}_1 = \frac{1}{100}\mathbf{v}, \mathbf{v}_2 = \frac{2}{100}\mathbf{v}, \ldots, \mathbf{v}_{100} = \mathbf{v}$. We will consider two measures to study the mesoscopic linearity:

$$\alpha = \frac{\|\nabla N(\mathbf{x}_0) - \nabla N(\mathbf{x}_0 + \mathbf{v}_i)\|}{\|\nabla N(\mathbf{x}_0)\|}, \quad \alpha_{\text{part}} = \frac{\|\nabla N(\mathbf{x}_0 + \mathbf{v}_{i+1}) - \nabla N(\mathbf{x}_0 + \mathbf{v}_i)\|}{\|\nabla N(\mathbf{x}_0)\|}.$$

Namely, $\alpha$ corresponds to Eq. (1) and measures the total change in gradient norm from $\mathbf{x}_0$ until $\mathbf{v}_i$, while $\alpha_{\text{part}}$ measure this change but only for consecutive steps. In Figure 2a and Figure 2b we plot $\alpha$ and $\alpha_{part}$ for all $i \in [100]$ for both adversarial and random step $\mathbf{v}$, respectively, for the ImageNet Swin-L [Liu et al., 2023] classifier and the CIFAR100 WRN-28-10 [Wang et al., 2023] classifier. The experiment details are in Appendix D.1.

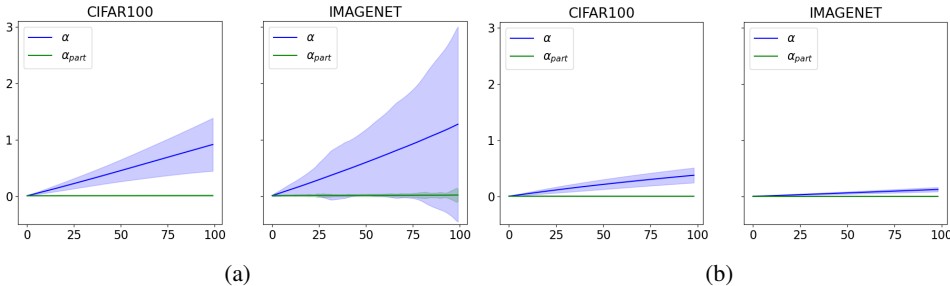

Figure 2: Measurement of **mesoscopic almost linearity** experimentally when taking a step $\mathbf{v}$ away from test image $x_0$ for CIFAR100 and ImageNet. The results are averaged over all the images in the test set, where (a) random step; and (b) Direction of the gradient (adversarial step).

It can be seen that for random directions (Figure 2b), both $\alpha$ and $\alpha_{\text{part}}$ are very small for both datasets, and with a very small variance. For adversarial directions (Figure 2a), $\alpha_{\text{part}}$ is still very small, while $\alpha$ is larger since it accumulates small deviations from linearity when moving further away from $\mathbf{x}_0$. We emphasize that by Eq. (1), mesoscopic almost linearity means that $\alpha = o_d(1)$, where $d$ is the dimension of the input. In other words, a larger input dimension should decrease, or at least not increase, the value of $\alpha$. In this experiment, the CIFAR100 and ImageNet datasets have very different input dimensions, namely, $d = 3072$ and $d = 150528$. We see that although the input dimension is almost 50 times larger for ImageNet compared to CIFAR100, the average value of $\alpha$ is nearly the same. This implies that mesoscopic almost linearity also happens in adversarial directions, and it would be interesting to further study how $\alpha$ changes when having even more drastic changes in the input dimension.

In our second experiment, we take a small adversarial step and compare the change of the logits of the output of the network and its linear approximation. In Figure 3, for each image $\mathbf{x}_0$, we find an adversarial direction $\mathbf{v}$, which we split into 100 equal parts similarly to the previous experiment. For a network $N$ we calculate its linear approximation $L$ at $\mathbf{x}_0$ using Taylor's expansion, and compare $N(\mathbf{x}_0 + \mathbf{v}_i)$ to $L(\mathbf{x}_0 + \mathbf{v}_i)$. Note that since $L$ is a linear function, $L(\mathbf{x}_0 + \mathbf{v}_i)$ is always a straight line when plotting the output for every $i = 1, \ldots, 100$. It can be seen that although $N(\mathbf{x}_0 + \mathbf{v}_i)$ is not necessarily linear, it does closely resemble the linear approximation, which suggests that almost linearity happens in the mesoscopic scale. We note that the logits from Figure 1b look less linear than in the figures below. We conjecture that this is because it is difficult to find an adversarial example for this image (and indeed, AutoAttack fails to do so). Hence, the network is less linear in those adversarial directions, and such examples are what causes the high variance in Figure 2a. This may require additional investigation, which we leave for future research.

## 5 Experiments

In this section, we compare the MALT attack (using a standard APGD attack) to the current state of the art AutoAttack. Our comparison is made to be compatible with RobustBench (Croce et al. [2020]), which is the standard benchmark for testing adversarial robustness. Namely, we consider attack on CIFAR-100 (Krizhevsky et al. [2009]) with an $\ell_\infty$ budget of $\epsilon = 8/255$ and on ImageNet (Deng et al. [2009]) with an $\ell_\infty$ budget of $\epsilon = 4/255$. We compare the two attacks on several robust models from the top RobustBench benchmark.

For MALT, we consider calculating the score for the $c = 100$ classes with the highest model's confidence and attacking the top $a = 9$ classes according to this score. This corresponds to the

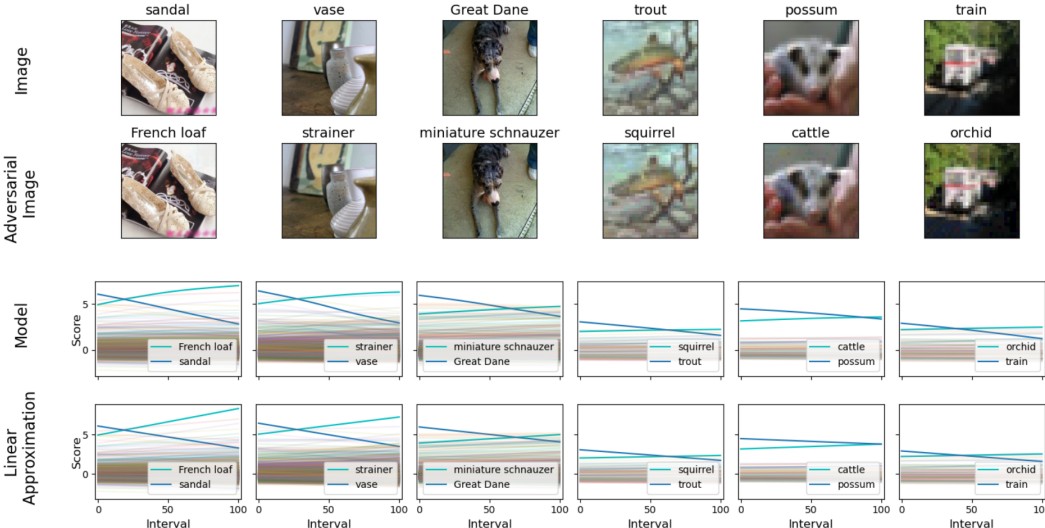

Figure 3: Empirical **mesoscopic almost linearity**: demonstrating the logits changes from an image $x_0$ to its adversarial example. In the third row, we plot the model's output logits changes, and in the bottom row are the results of the linear approximation of the model at $x_0$.

targeted attacks in AutoAttack, which attempt to attack the top 9 classes according to the model's confidence. All the hyperparameters of APGD and the other attacks used in AutoAttack are set to their default values. Full details for all experiments in this section are in Appendix D.

In Table 1 (CIFAR-100) and Table 2 (ImageNet), we present the results of the experiments. Note that MALT is able to attack more images than AutoAttack across all robust models and for both datasets. Also, there is an inclusion of successful attacks in the sense that for every image that AutoAttack successfully attacks, MALT also succeeds. Thus, the improvement contains only images that AutoAttack fails on. Note that in Table 2, the robust and clean accuracy is not exactly equal to those reported on the RobustBench website since newer versions of the Python libraries in use give slightly different results.

Table 1: **CIFAR100 -** $L_\infty$ robust accuracy *(lower is better)*, comparing MALT and AutoAttack, which is the current state of the art.

| MODEL | | ROBUSTNESS | | | |
|---|---|---|---|---|---|
| | ACC. | MALT | SOTA | DIFF | SPEED-UP |
| WRN-28-10 [WANG ET AL., 2023] | 72.58% | 38.79% | 38.83% | −0.04% | × 3.36 ±0.18 |
| WRN-70-16 [WANG ET AL., 2023] | 75.22% | 42.66% | 42.67% | −0.01% | × 3.87 ±0.08 |
| WRN-28-10 [CUI ET AL., 2023] | 73.83% | 39.18% | 39.18% | 0% | × 3.43 ±0.08 |
| WRN-70-16 [GOWAL ET AL., 2020] | 69.15% | 36.81% | 36.88% | −0.07% | × 3.42 ±0.09 |

Table 2: **ImageNet -** $L_\infty$ robust accuracy *(lower is better)*, comparing MALT and AutoAttack, which is the current state of the art.

| MODEL | | ROBUSTNESS | | | |
|---|---|---|---|---|---|
| | ACC. | MALT | SOTA | DIFF. | SPEED-UP |
| SWIN-L [LIU ET AL., 2023] | 79.18% | 59.84% | 59.90% | −0.06% | × 5.18 ±0.04 |
| CONVNEXT-L [LIU ET AL., 2023] | 78.20% | 58.82% | 58.88% | −0.06% | × 5.22 ±0.1 |
| CONVNEXT-L+ [SINGH ET AL., 2024] | 77.02% | 57.94% | 57.96% | −0.02% | × 4.86 ±0.06 |
| SWIN-B [LIU ET AL., 2023] | 76.22% | 56.54% | 56.56% | −0.02% | × 5.02 ±0.03 |
| CONVNEXT-B+ [SINGH ET AL., 2024] | 76.00% | 56.48% | 56.52% | −0.04% | × 5.00 ±0.07 |

**Attack Running Time** For the speed-up column, we first sampled uniformly from the test dataset five batches of size 200 for ImageNet and 400 for CIFAR100, which corresponds to $5 \times 4\%$ of the entire test set. We timed both MALT and AutoAttack, when running on the exact same GPU chip and on the same samples, we present the mean and variance of these experiments for each robust model. On the ImageNet dataset there is on average a five times speed up, while for CIFAR100 it is 3.5 times on average. The lower rate can be explained by the higher attack success rates for CIFAR100. In other words, the complexity analysis from Section 3 is of the worst case, and since fewer test examples pass through all four different attacks in AutoAttack, the running time is lower.

**Integrating MALT with different attacks** Our experiments found that MALT operates well across datasets and models with the standard and fast APGD attack. We also integrated MALT with the targeted FAB attack (Croce and Hein [2020a]) as well as with the APGD attack with the CE loss, on the Swin-L robust model (Liu et al. [2023]). Both attacks achieved a worse results of $60.64\%$ for the FAB attack and $60.52\%$ for the APGD attack with CE loss, compared to the superior $59.84\%$ robust accuracy using APGD with the DLR loss.

## 5.1 Targeting analysis

In this subsection, we analyze the targeting mechanism of MALT and whether other targeting techniques could have improved it. In Figure 4, we show how the successful attacks are distributed according to the score given either by MALT or the naive targeting (i.e., according to the model's confidence). It is evident that the score provided by MALT leans much more towards the $top - 1$ than naive targeting. This indicates that the score provided by MALT has a higher correlation with successful attacks than the naive targeting method. Also, since MALT successfully attacks more images for the top 9 targets (achieving robust accuracy of $59.84\%$, compared to $59.94\%$ for naive targeting), the sum over the columns of MALT attacks is larger than that of the naive targeting.

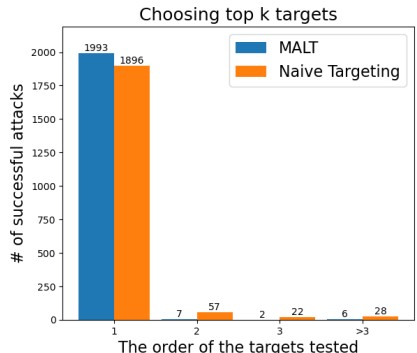

Figure 4: Comparing targeting methods for Liu et al. [2023] SOTA model: The number of successful attacks for each target order by two targeting methods: In blue, we use MALT targeting and APGD, and in orange, we compare to APGD with top logits targeting performed in AutoAttack.

Finally, we test whether the choice of $c = 100$ is enough, namely calculating the score for MALT only for the top 100 classes sorted by the confidence level of the model. To this end, we ran MALT with $c = 1000$ for the ImageNet test dataset on the Swin-L robust network. Note that the number of gradient calculations for each test image has increased by a factor of more than two. This attack reached a robust accuracy of $59.84\%$, *exactly the same as $c = 100$*. This means that calculating the score for the top 100 classes is enough to find the top classes to attack. It may be possible to optimize this hyperparameter even more, although the benefit in running time will be negligible.

## 6 Conclusions

In this paper we present MALT, an adversarial attack which is based on a targeting method that assumes almost linearity in the mesoscopic scale. MALT wins over the current state of the art adversarial attack AutoAttack on several robust models, and for both CIFAR100 and ImageNet, while also speeding up the runtime by more than *five times*. We also present theoretical and empirical evidence that our almost linearity assumption is applied to neural network, in the mesoscopic scale where adversarial examples exist.

There are several future research directions that we think are interesting. First, it would be interesting to further study the mesoscopic almost linearity property of neural networks. In particular, whether this property is affected by (or affecting) the robustness of the network to adversarial perturbations, and the perturbation's transferability capabilities. Theoretically, it would be interesting to extend our

analysis to deeper networks. As for the applied results, a good future direction would be to conduct a more extensive empirical study of MALT on more robust models and datasets, even beyond the scope of the RobustBench benchmark. Finally, it would be interesting to understand whether there exists a better targeting method to find adversarial classes, e.g., using a second-order approximation of the network.

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

# A  Additional MALT Targeting Examples

Following Section 3, to further present the performance of the MALT targeting method, we add more examples of images on which AutoAttack fails and MALT succeeds. We present such images from the CIFAR100 and the ImageNet datasets in Figure 6 and Figure 5 respectively. As in the examples found in Section 3, Figure 1, we present two attacks on each image - the naively targeted APGD and the MALT attack using APGD.

**MALT and APGD**  For all experiments, for the MALT attack, we used $a = 9$ and $c = 100$, and the default hyperparameters of the attack APGD adapted from the official AutoAttack git repository.[4] Note that we used the APGD attack with the given default DLR loss.

In the upper row, for each image, one can see a failed APGD attack toward the class with the model's second-best confidence level. On the bottom row, we present our successful attack - a successful APGD attack toward a target found using the MALT method. The original image is on the left, and the perturbed image is on the right. Between them, we present the change in the model's output logits, starting from the original image and ending in the perturbed one.

Note that in the CIFAR100 dataset experiments, the APGD attack gets random errors due to randomization. We count both here in Figure 6 and in Section 5 only images that MALT targets to class **not within** the naive top nine logits, therefore a clear advantage to MALT, regardless of these randomization-resulted errors.

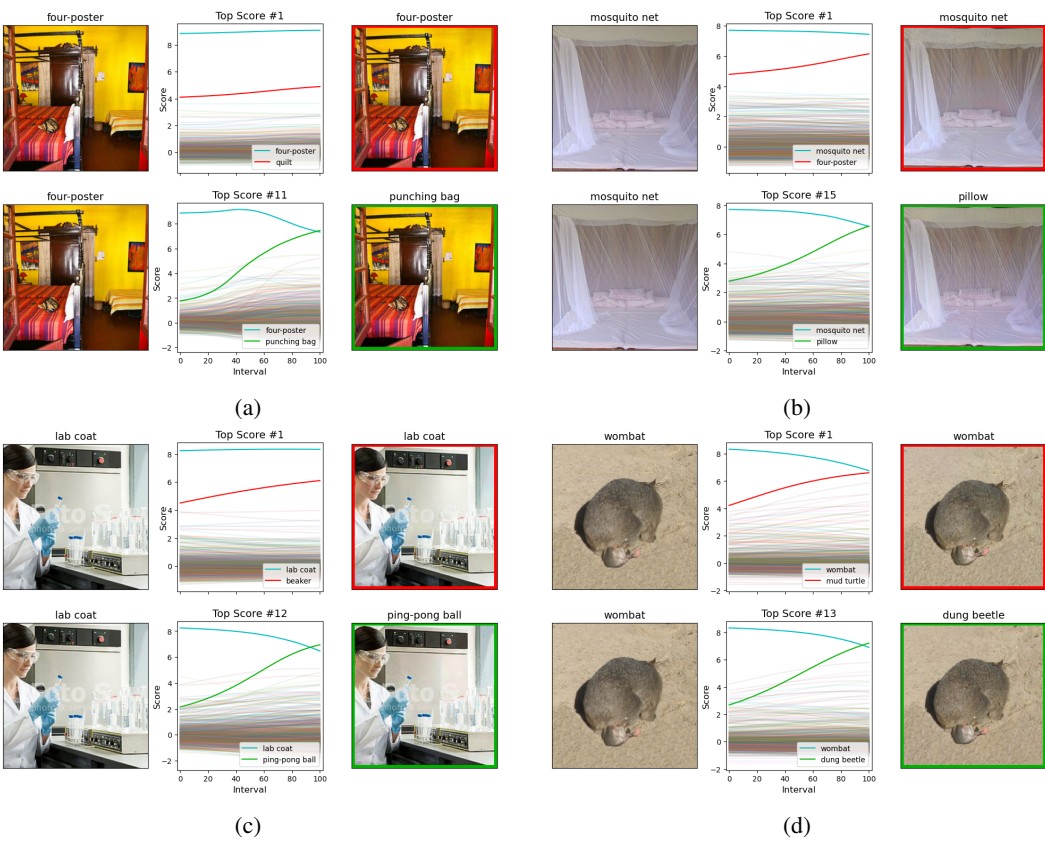

[4]https://github.com/fra31/auto-attack

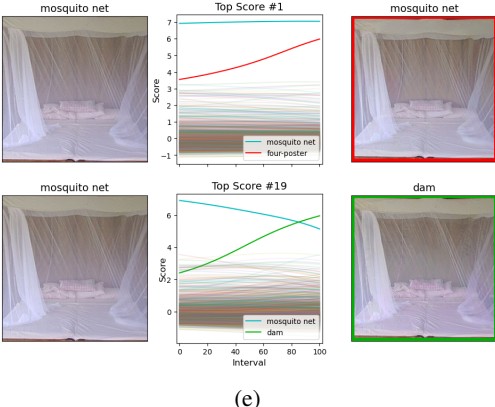

(e)

Figure 5: Additional examples of images from the ImageNet dataset that AutoAttack fails to attack while MALT succeeds. The top row shows an APGD attack on the target class with the highest logit, and the bottom row shows an APGD attack on the class that MALT finds and succeeds. (a) and (b) examples from Swin-L [Liu et al., 2023] network. (c) through (e) are from ConvNext-L [Liu et al., 2023] network. The images are shown before and after the attack, and the change in logits is presented in the middle column.

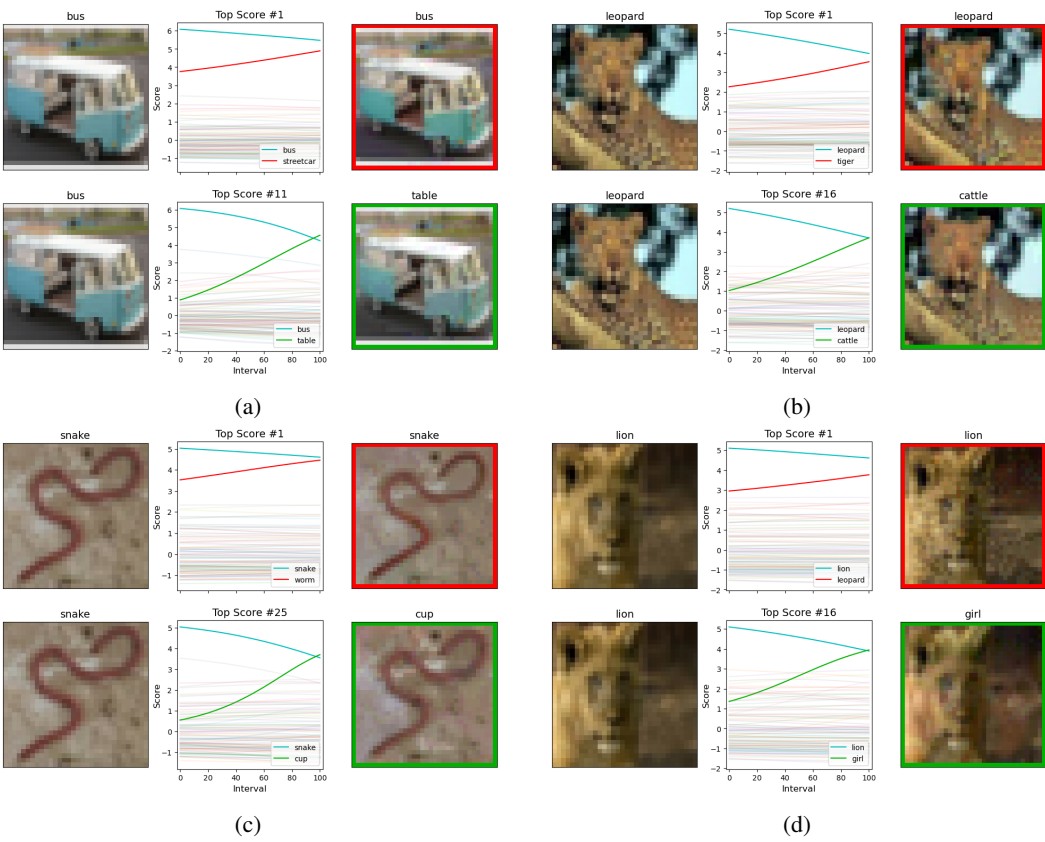

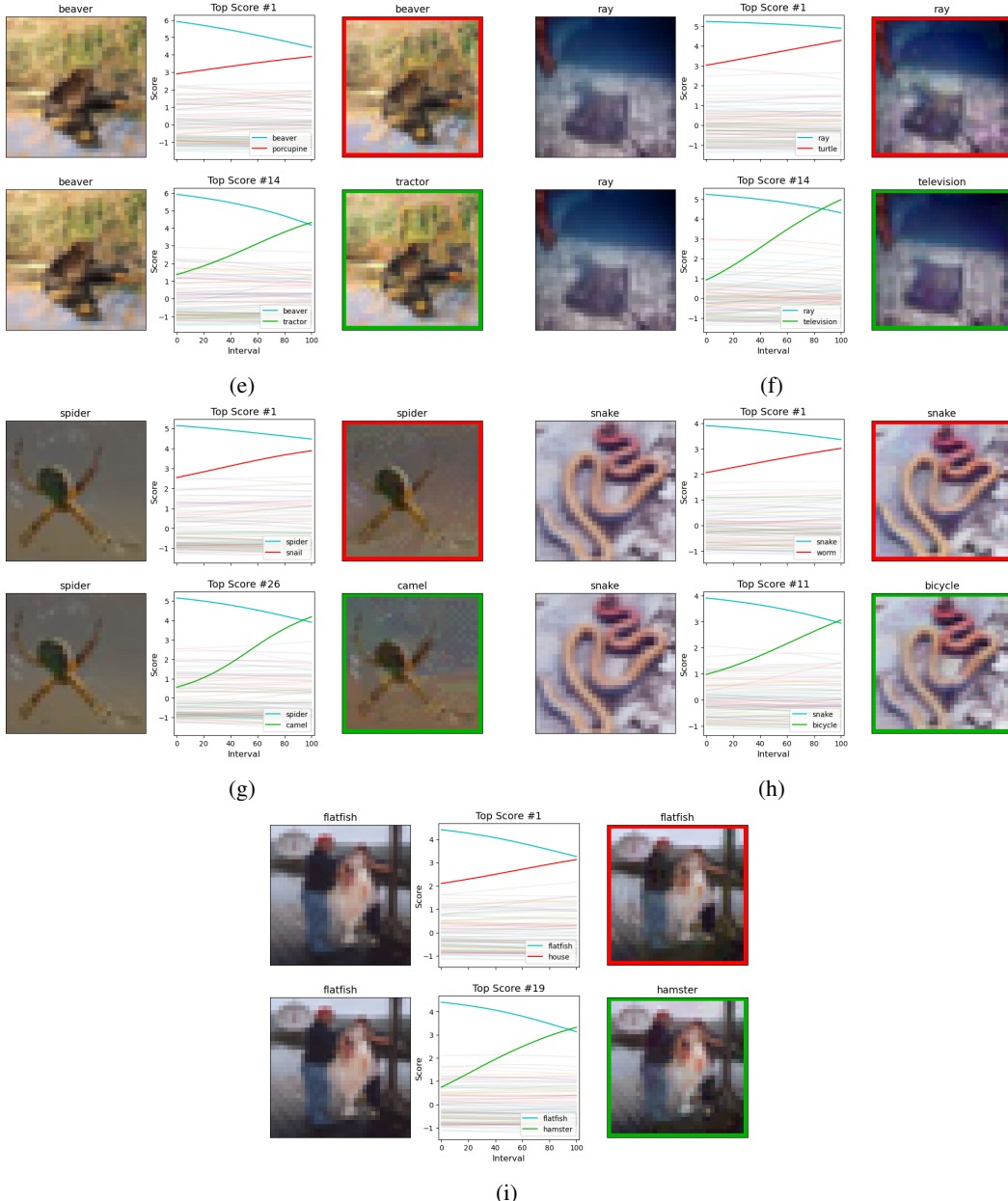

Figure 6: Additional examples of images from the CIFAR100 dataset that AutoAttack fails to attack while MALT succeeds. The top row shows an APGD attack on the target class with the highest logit, and the bottom row shows an APGD attack on the class that MALT finds and succeeds. (a) through (g) examples from WRN-70-16 [Gowal et al., 2020] network. (h) is from WRN-28-10 [Wang et al., 2023] network. (i) is from WRN-70-16 [Wang et al., 2023] network. The images are shown before and after the attack, and the change in logits is presented in the middle column.

# B  Proofs from Section 3

*Proof of Lemma 3.1.* Consider finding the smallest adversarial perturbation $\mathbf{z}_i$ for $\mathbf{x}_0$ where the $i$-th output of $F$ is larger than the $\ell$-th output of $F$. That is, we want to find:

$$\min_{\mathbf{z}} \|\mathbf{z}\|^2 \text{ s.t. } F_i(\mathbf{x}_0 + \mathbf{z}_i) - F_\ell(\mathbf{x}_0 + \mathbf{z}_i) > 0 \tag{2}$$

We have that: $F_i(\mathbf{x}_0 + \mathbf{z}_i) - F_\ell(\mathbf{x}_0 + \mathbf{z}_i) = \langle \mathbf{w}_i, \mathbf{x}_0 + \mathbf{z}_i \rangle - \langle \mathbf{w}_\ell, \mathbf{x}_0 + \mathbf{z}_i \rangle$. Hence, we get that Eq. (2) is a constraint minimization problem with a single linear constraint. The solution to this problem is given by the direction $\mathbf{z}_i \propto \mathbf{w}_i - \mathbf{w}_\ell$. We can write $\mathbf{z}_i = \epsilon_i(\mathbf{w}_i - \mathbf{w}_\ell)$ for $\epsilon_i \in \mathbb{R}$ which will be determined later, then we have that:

$$\begin{aligned}
F_i(\mathbf{x}_0 + \mathbf{z}_i) - F_\ell(\mathbf{x}_0 + \mathbf{z}_i) &= \langle \mathbf{w}_i, \mathbf{x}_0 + \mathbf{z}_i \rangle - \langle \mathbf{w}_\ell, \mathbf{x}_0 + \mathbf{z}_i \rangle \\
&= \langle \mathbf{w}_i, \mathbf{x}_0 + \epsilon_i(\mathbf{w}_i - \mathbf{w}_\ell) \rangle - \langle \mathbf{w}_\ell, \mathbf{x}_0 + \epsilon_i(\mathbf{w}_i - \mathbf{w}_\ell) \rangle \\
&= \langle \mathbf{w}_i - \mathbf{w}_\ell, \mathbf{x}_0 \rangle + \epsilon_i \|\mathbf{w}_i - \mathbf{w}_\ell\|^2 \ .
\end{aligned}$$

That is, for $\epsilon_i = \frac{\langle \mathbf{w}_i - \mathbf{w}_\ell, \mathbf{x}_0 \rangle}{\|\mathbf{w}_i - \mathbf{w}_\ell\|^2}$ and $\mathbf{z}_i = \epsilon_i(\mathbf{w}_i - \mathbf{w}_\ell)$ we have that $F_i(\mathbf{x}_0 + \mathbf{z}_i) = F_\ell(\mathbf{x}_0 + \mathbf{z}_i)$. Note that $\|\mathbf{z}_i\| = \frac{\langle \mathbf{w}_i - \mathbf{w}_\ell, \mathbf{x}_0 \rangle}{\|\mathbf{w}_i - \mathbf{w}_\ell\|}$, we omit the absolute value on the nominator since the constraint in Eq. (2) is that it is positive. Finally, finding $\arg\min_i \|\mathbf{z}_i\| = \arg\min_i \frac{\langle \mathbf{w}_i - \mathbf{w}_\ell, \mathbf{x}_0 \rangle}{\|\mathbf{w}_i - \mathbf{w}_\ell\|}$ provides the class index with the smallest norm adversarial perturbation. $\square$

# C  Proofs from Section 4

## C.1  Proof of Theorem 4.1

*Proof.* Using Lemma C.1 we can assume w.l.o.g that $P = \text{span}\{\mathbf{e}_1, \ldots, \mathbf{e}_{d-\ell}\}$. Denote by $\hat{\mathbf{w}} := \Pi_{P^\perp}(\mathbf{w})$ the projection of $\mathbf{w}$ on the subspace $P^\perp$. By Lemma C.2 we get that the weights $\hat{\mathbf{w}}_i$ did not change from their initial value for any $i \in \{1, \ldots, m\}$.

We will now calculate the gradient of the network w.r.t the data. For ease of notations we drop the $\mathbf{w}_{1:m}$ as the input of the network.

$$\nabla_{\mathbf{x}} N(\mathbf{x}) = \sum_{i=1}^{m} u_i \mathbf{w}_i \sigma'(\mathbf{w}_i^\top \mathbf{x})$$

Thus, we want to bound the following:

$$\begin{aligned}
&\left\| \Pi_{P^\perp} \left( \sum_{i=1}^{m} u_i \mathbf{w}_i \sigma'(\mathbf{w}_i^\top \mathbf{x}) - \sum_{i=1}^{m} u_i \mathbf{w}_i \sigma'(\mathbf{w}_i^\top (\mathbf{x} + \mathbf{v})) \right) \right\| \\
&\leq \left\| \sum_{i=1}^{m} u_i \hat{\mathbf{w}}_i \left( \sigma'(\mathbf{w}_i^\top \mathbf{x}) - \sigma'(\mathbf{w}_i^\top \mathbf{x} + \mathbf{v}) \right) \right\|
\end{aligned}$$

Using that $\|\mathbf{w}\| = \sup_{\mathbf{u} \in \mathbb{S}^{d-1}} \mathbf{w}^\top \mathbf{u}$, it is enough to bound the following for any $\mathbf{u} \in \mathbb{S}^{d-1}$:

$$\sum_{i=1}^{m} u_i \hat{\mathbf{w}}_i^\top \mathbf{u} \left( \sigma'(\mathbf{w}_i^\top \mathbf{x}) - \sigma'(\mathbf{w}_i^\top \mathbf{x} + \mathbf{v}) \right) \ . \tag{3}$$

We will now use Bernstein inequality on the above sum. Denote $X_i = u_i \hat{\mathbf{w}}_i^\top \mathbf{u} \left( \sigma'(\mathbf{w}_i^\top \mathbf{x}) - \sigma'(\mathbf{w}_i^\top \mathbf{x} + \mathbf{v}) \right)$, then we have that:

$$\begin{aligned}
\mathbb{E}[|X_i|^q] &\leq \frac{L^q}{m^{q/2}} \mathbb{E}\left[ |\hat{\mathbf{w}}^\top \mathbf{u}| \cdot |\hat{\mathbf{w}}^\top \mathbf{v}| \right] = \frac{L^q}{m^{q/2}} \sqrt{\mathbb{E}[|\hat{\mathbf{w}}^\top \mathbf{u}|]^{2q} \cdot \mathbb{E}[|\hat{\mathbf{w}}^\top \mathbf{v}|]^{2q}} \\
&\leq \frac{(LR)^q}{(d-\ell)^q m^{q/2}} \mathbb{E}_{Y \sim N(0,1)}[|Y|^{2q}] \leq \frac{(LR)^q}{(d-\ell)^q m^{q/2}} (2q-1)!! \leq \frac{q!}{2} \cdot \frac{(LR)^q}{(d-\ell)^q m^{q/2}} \ ,
\end{aligned}$$

where we used that $\sigma$ is $L$-smooth, and the assumption on the initialization of $\mathbf{w}_i$ and $u_i$. Note that the above is true for every $\mathbf{u} \in \mathbb{S}^{d-1}$, hence by Bernstein inequality with $c = \frac{LR}{(d-\ell)\sqrt{m}}$ we have w.p $> 1 - \delta$ over the initialization:

$$\left\| \sum_{i=1}^m u_i \hat{\mathbf{w}}_i \left( \sigma'(\mathbf{w}_i^\top \mathbf{x}) - \sigma'(\mathbf{w}_i^\top \mathbf{x} + \mathbf{v}) \right) \right\| \leq \frac{LR}{d-\ell} \sqrt{\log\left(\frac{1}{\delta}\right)} \left( 1 + \sqrt{\frac{\log\left(\frac{1}{\delta}\right)}{m}} \right) . \qquad (4)$$

Denote by $\Omega := \{ (\mathbf{u}, \mathbf{v}) : \mathbf{u} \in \mathbb{S}^{d-1}, \|\mathbf{v}\| \leq R, \mathbf{v}, \mathbf{u} \in P^\perp, \}$, we will now bound Eq. (3) uniformly over $\Omega$. Denote by $\Phi(\mathbf{u}, \mathbf{v}) := \sum_{i=1}^m u_i \hat{\mathbf{w}}_i^\top \mathbf{u} \left( \sigma'(\mathbf{w}_i^\top \mathbf{x}) - \sigma'(\mathbf{w}_i^\top \mathbf{x} + \mathbf{v}) \right)$. We define an $\epsilon$-net over $\Omega$ for $\epsilon$ to be chosen later, denote this net by $M_\epsilon$. The size of the $M_\epsilon$ is at most $\left( \frac{10R}{\epsilon} \right)^{d-\ell}$, and we have that:

$$\sup_{(\mathbf{u},\mathbf{v}) \in \Omega} \Phi(\mathbf{u}, \mathbf{v}) \leq \sup_{(\mathbf{u},\mathbf{v}) \in M_\epsilon} \Phi(\mathbf{u}, \mathbf{v}) + \sup_{(\mathbf{u},\mathbf{v}),(\mathbf{u}',\mathbf{v}') \in \Omega, \|\mathbf{u}+\mathbf{u}'\|+\|\mathbf{v}-\mathbf{v}'\| \leq \epsilon} |\Phi(\mathbf{u}, \mathbf{v}) - \Phi(\mathbf{u}', \mathbf{v}')| .$$
$$(5)$$

To bound the first term of Eq. (5) we use a union bound over all of $M_\epsilon$, and by Eq. (4) we have w.p $> 1 - \delta$:

$$\sup_{(\mathbf{u},\mathbf{v}) \in M_\epsilon} \Phi(\mathbf{u}, \mathbf{v}) \leq \frac{LR}{d-\ell} \sqrt{(d-\ell)\log\left(\frac{1}{\epsilon}\right) + \log\left(\frac{1}{\delta}\right)} \left( 1 + \sqrt{\frac{(d-\ell)\log\left(\frac{1}{\epsilon}\right) + \log\left(\frac{1}{\delta}\right)}{m}} \right) .$$
$$(6)$$

For the second term in Eq. (5), note that for any $\mathbf{u}, \mathbf{u}', \mathbf{v}, \mathbf{v}'$:

$$|\Phi(\mathbf{u}, \mathbf{v}) - \Phi(\mathbf{u}', \mathbf{v})| \leq \frac{L\|\mathbf{u} - \mathbf{u}'\|}{\sqrt{m}} \sum_{i=1}^m \|\hat{\mathbf{w}}_i\|^2$$

$$|\Phi(\mathbf{u}, \mathbf{v}) - \Phi(\mathbf{u}, \mathbf{v}')| \leq \frac{LR\|\mathbf{v} - \mathbf{v}'\|}{\sqrt{m}} \sum_{i=1}^m \|\hat{\mathbf{w}}_i\|^2$$

Note that $\sum_{i=1}^m \|\hat{\mathbf{w}}_i\|^2$ is a scaled chi-squared distribution with $m \cdot d$ degrees of freedom. By Lemma 1 in Laurent and Massart [2000] we have w.p $> 1 - \delta$ that:

$$\sum_{i=1}^m \|\hat{\mathbf{w}}_i\|^2 \leq m + 2\sqrt{\frac{m \log\left(\frac{1}{\delta}\right)}{d-\ell}} + \frac{2\log\left(\frac{1}{\delta}\right)}{d-\ell} .$$

Combining the above, and taking $\epsilon = \frac{1}{m}$ we get:

$$\sup_{(\mathbf{u},\mathbf{v}),(\mathbf{u}',\mathbf{v}') \in \Omega, \|\mathbf{u}+\mathbf{u}'\|+\|\mathbf{v}-\mathbf{v}'\| \leq \frac{1}{m}} |\Phi(\mathbf{u}, \mathbf{v}) - \Phi(\mathbf{u}', \mathbf{v}')| \leq LR \left( \frac{1}{\sqrt{m}} + 2\sqrt{\frac{\log\left(\frac{1}{\delta}\right)}{d-\ell}} + \frac{2\log\left(\frac{1}{\delta}\right)}{(d-\ell)\sqrt{m}} \right)$$

Plugging this into Eq. (5), while choosing $\delta' = \frac{\delta}{2}$ finishes the proof $\qquad \square$

### C.2   Proof of Theorem 4.2

*Proof.* Denote by $\hat{\mathbf{w}} := \Pi_{P^\perp}(\mathbf{w})$ the projection of $\mathbf{w}$ on the subspace $P^\perp$. We have that:

$$\|\Pi_{P^\perp}(\nabla_{\mathbf{x}} N(\mathbf{x}))\| = \left\| \sum_{i=1}^m u_i \hat{\mathbf{w}}_i \sigma'(\mathbf{w}_i^\top \mathbf{x}) \right\| \geq \frac{\beta}{\sqrt{m}} \left\| \sum_{i=1}^m \hat{\mathbf{w}}_i \right\| .$$

Using Lemma C.1 we can assume w.l.o.g that $P = \text{span}\{\mathbf{e}_1, \ldots, \mathbf{e}_{d-\ell}\}$. By Lemma C.2 we get that the weights $\hat{\mathbf{w}}_i$ did not change from their initial value for any $i \in \{1, \ldots, m\}$. Hence, $\hat{\mathbf{w}}_i \sim N\left(0, \frac{1}{d}I\right)$ for every $i$, then $\sum_{i=1}^m \hat{\mathbf{w}}_i \sim N\left(0, \frac{m}{d}I\right)$. By Lemma 1 in Laurent and Massart [2000] w.p $> 1 - \delta$ we have that:

$$\left\| \sum_{i=1}^m \hat{\mathbf{w}}_i \right\| \geq \sqrt{m - 2\sqrt{\frac{m \log\left(\frac{1}{\delta}\right)}{d}}} .$$

Combining the two displayed equations yields:

$$\|\Pi_{P^\perp}(\nabla_{\mathbf{x}} N(\mathbf{x}))\| \geq \beta \cdot \sqrt{1 - 2\sqrt{\frac{\log\left(\frac{1}{\delta}\right)}{d}}} \ .$$

$\square$

## C.3 Additional Lemmas

The following the lemmas are taken from Melamed et al. [2023], and used to assume w.l.o.g that $P = \text{span}\{\mathbf{e}_1, \ldots, \mathbf{e}_{d-\ell}\}$, and that the weights of the neurons projected on $P^\perp$ do not change during training. We provide them here for completeness[5].

**Lemma C.1.** *[Theorem A.1 from Melamed et al. [2023]] Let $P \subseteq \mathbb{R}^d$ be a subspace of dimension $d - \ell$, and let $M = \text{span}\{\mathbf{e}_1, \ldots, \mathbf{e}_{d-\ell}\}$. Let $R$ be an orthogonal matrix such that $R \cdot P = M$, let $X \subseteq P$ be a training dataset and let $X_R = \{R \cdot \mathbf{x} : \mathbf{x} \in X\}$. Assume we train a neural network $N(\mathbf{x}) = \sum_{i=1}^{m} u_i \sigma(\mathbf{w}_i^\top \mathbf{x})$ as explained in Section 4, and denote by $N^X$ and $N^{X_R}$ the network trained on $X$ and $X_R$ respectively for the same number of iterations. Let $\mathbf{x}_1, \mathbf{x}_2 \in P$, then we have:*

1. *W.p. $p$ (over the initialization) we have $\left\|\Pi_{P^\perp}\left(\frac{\partial N^X(\mathbf{x}_1)}{\partial \mathbf{x}}\right)\right\| \geq c$ (resp. $\leq c$) for some $c \in \mathbb{R}$, iff w.p. $p$ also $\left\|\Pi_{M^\perp}\left(\frac{\partial N^{X_R}(R\mathbf{x}_1)}{\partial \mathbf{x}}\right)\right\| \geq c$ (resp. $\leq c$).*

2. *W.p. $p$ (over the initialization) we have $\left\|\Pi_{P^\perp}\left(\frac{\partial N^X(\mathbf{x}_1)}{\partial \mathbf{x}}\right) - \Pi_{P^\perp}\left(\frac{\partial N^X(\mathbf{x}_2)}{\partial \mathbf{x}}\right)\right\| \geq c$ (resp. $\leq c$) for some $c \in \mathbb{R}$, iff w.p. $p$ also $\left\|\Pi_{M^\perp}\left(\frac{\partial N^{X_R}(R\mathbf{x}_1)}{\partial \mathbf{x}}\right) - \Pi_{M^\perp}\left(\frac{\partial N^{X_R}(R\mathbf{x}_2)}{\partial \mathbf{x}}\right)\right\| \geq c$ (resp. $\leq c$).*

**Lemma C.2** (Theorem A.2 from Melamed et al. [2023]). *Let $M = \text{span}\{\mathbf{e}_1, \ldots, \mathbf{e}_{d-\ell}\}$. Assume we train a neural network $N(\mathbf{x}, \mathbf{w}_{1:m}) := \sum_{i=1}^{m} u_i \sigma(\mathbf{w}_i^\top x)$ as explained in Section 4 (where $\mathbf{w}_{1:m} = (w_1, \ldots, w_m)$). Denote by $\hat{\mathbf{w}} := \Pi_{M^\perp}(\mathbf{w})$ for $\mathbf{w} \in \mathbb{R}^d$, then after training, for each $i \in [m]$, $\hat{\mathbf{w}}_i$ did not change from their initial value.*

The following is Bernstein lemma, adapted from the phrasing of Theorem 3 in Bubeck et al. [2021].

**Lemma C.3.** *Let $X_i$ for $i = 1, \ldots, m$ be i.i.d random variables with zero mean, such that there exists $c > 0$ that for all integers $q \geq 2$ we have:*

$$\mathbb{E}[|X_i|^q] \leq \frac{q! c^q}{2}$$

*Then, w.p $> 1 - \delta$ we have that:*

$$\sum_{i=1}^{m} X_i \leq \sqrt{2c^2 k \log\left(\frac{1}{\delta}\right)} + c \log\left(\frac{1}{\delta}\right)$$

## C.4 Assumption on the theorems

Here we discuss the different assumptions made in the formal proofs, and where they are used. We separate between the assumption on the activation, and the assumptions on the parameters made in Corollary 4.1.

**Assumptions on the activation.** The assumption that the activation is $L$-smooth is used in the proof Theorem 4.1 to bound the difference between the gradient at different points. Bubeck and Sellke [2021] generalize this result to the ReLU activation, however they assume that the data point are drawn randomly from a Gaussian. The reason for such an assumption is to bound the number of neurons which change the sign for ReLU networks. In our case, since we don't assume random data points, the number of neurons that change sign depends on the data, and the neurons $\mathbf{w}_i$, which during

---

[5]The second item of Lemma C.1 does not appear as is in Melamed et al. [2023], but proved in the exact same way as first item.

training with gradient descent also become dependent on the data points. Thus, the generalization from Bubeck and Sellke [2021] is not applicable in our case.

The assumption that the derivative is lower bounded by some constant is used in the proof of Theorem 4.2. There, the norm of the gradient can be bounded more generally by the term $\sum_{i=1}^{m} \sigma'(\mathbf{w}_i^\top \mathbf{x})^2$. Again, note that each neuron $\mathbf{w}_i$ depends on the data that the network trained on. Hence, if $\sigma'$ is not lower bounded, even at a single point, then $\sigma'(\mathbf{w}_i^\top \mathbf{x})^2$ can be very close to zero for every $i$, and thus the bound becomes vacuous.

Both assumptions can be relaxed by adding more assumptions on the data. For example, if we assume that the data is drawn randomly from a Gaussian, then we could use similar arguments to those used in Bubeck et al. [2021] to generalize these results. Another possible direction is to assume over-parameterization, i.e. the number of neurons is much larger than the number of samples. In this case, it is possible to analyze these results in the NTK regime Jacot et al. [2018].

**Assumptions in Corollary 4.1.**  Here, besides the assumption on the activation, we additionally have assumptions on the parameters of the problem. We explain in details the interpretation of each assumption:

1. $d - \ell = \Omega(d)$. This assumption means that the dimension of the orthogonal subspace to $P$ is large enough. Such an assumption is also needed in Melamed et al. [2023], Haldar et al. [2024], since the analysis of adversarial perturbation is made inside the subspace $P^\perp$, which needs to be large enough.

2. $R = o(\sqrt{d - \ell})$. This just means that the perturbation is not too large. This is equivalent to saying that we focus on the mesoscopic scale, i.e. where the network is non-linear, but still behaves as if it is almost linear.

3. $m = \Omega(d - \ell)$. This assumption is used since the bound on the gradient depends both on the input dimension, and the number of neurons. This assumption can be interpreted as if the number of neurons cannot be much smaller than the dimension of the subspace $P^\perp$ on which the adversarial perturbations exist.

# D   Experimental Details

## D.1   Experiments from Section 4.2

In Figure 2, for an image $\mathbf{x}_0$ in the test dataset, we create a perturbation vector $\mathbf{v}$ of norm $\epsilon = 8/255$ or $\epsilon = 4/255$ for CIFAR100 or ImageNet datasets respectively, with respect to the $L_\infty$ norm. The models considered are the ImageNet Swin-L [Liu et al., 2023] classifier and the CIFAR100 WRN-28-10 [Wang et al., 2023] classifier.

We consider two different choices of such $\mathbf{v}$:

1. Random direction: we start from a random $\mathbf{v} \sim \mathcal{U}\left(\{\pm\epsilon\}\right)^d$. This randomization is common for adversarial attacks.

2. Gradient direction: we calculate the gradient of the network w.r.t. the input at $\mathbf{x}_0$, $\nabla_{\mathbf{x}} N(\mathbf{x}_0)$, and normalize it to be of norm $\epsilon$.

In both cases, we then truncate $\mathbf{v}$ to the input domain so that $\mathbf{x}_0 + \mathbf{v}$ would not exceed $\pm 1$ in each coordinate.

We divide $\mathbf{v}$ to 100 equal parts, namely $\mathbf{v}_1 = \frac{1}{100}\mathbf{v}, \mathbf{v}_2 = \frac{2}{100}\mathbf{v}, \ldots, \mathbf{v}_{100} = \mathbf{v}$. For each $i$, we calculate:

$$\alpha = \frac{\|\nabla N(\mathbf{x}_0) - \nabla N(\mathbf{x}_0 + \mathbf{v}_i)\|}{\|\nabla N(\mathbf{x}_0)\|}, \quad \alpha_{\text{part}} = \frac{\|\nabla N(\mathbf{x}_0 + \mathbf{v}_{i+1}) - \nabla N(\mathbf{x}_0 + \mathbf{v}_i)\|}{\|\nabla N(\mathbf{x}_0)\|}.$$

For each $i$, we take an average and a mean over all the test datasets, viewed in Figure 2.

### D.2 Experiments from Section 5

We use the same $\epsilon$ budget for all experiments - $8/255$ and $4/255$ in $L_\infty$ norm for CIFAR100 and ImageNet datasets, respectively.

#### D.2.1 Robust Accuracy Experiment Details

**AutoAttack** For all experiments, if needed, we run AutoAttack with its default hyperparameters taken from the RobustBench git repository.[6] For the CIFAR100 dataset, we took the clean and robust accuracy results directly from the RobustBench website. For the ImageNet dataset, we noticed that even the clean accuracy of the networks changes when using different versions of the Pytorch library. This clearly affects the robust accuracy of the networks since misclassified test images are considered their own adversarial examples. Therefore, we re-calculate the clean and robust accuracy for all considered networks, presenting the results of our execution.

**MALT and APGD** For all experiments, for the MALT targeting method, we used $a = 9$ and $c = 100$, and the default hyperparameters of the APGD attack, adapted from the official AutoAttack git repository.[7] Note that we used the APGD attack with the given default DLR loss.

Our attack uses an APGD attack within it; therefore, it suffers from a common issue of randomization in the CIFAR100 executions (also mentioned in Appendix A). Consequently, we ran these experiments five times for each model examined and took the best value to avoid these randomization-resulted errors.

**General execution details** For the CIFAR100 robust classifiers, we execute the attack in batches of 200 samples each and 50 sample batches for the ImageNet classifiers. All the weights for all the classifiers were downloaded directly from the RobustBench git repository. All experiments were done using a GPU Tesla V-100, 16GB.

#### D.2.2 Running Time Experiment Details

In this experiment presented in the rightmost column of Table 1 and Table 2, we sample five independent image subsets for each dataset, consisting of 400 and 200 images from CIFAR100 and ImageNet datasets, respectively. For each sample, we run both AutoAttack and our attack on the entire sample and measure the attack running time. Later, for each network, we calculate the mean and standard deviation over all five samples. The CIFAR100 attacks ran all on 2 GPUs of Tesla V-100, 16GB. The ImageNet attacks ran on 3 GPUs of Tesla V-100, 16GB. Of course, for each network, both attacks were executed one after the other using the exact same computing power.

#### D.2.3 MALT and FAB Details

For this experiment, we take the same $a = 9$ and $c = 100$ hyperparameters for MALT and take FAB implementation from the official AutoAttack git repository, with all its default hyperparameters.

#### D.2.4 Targeting Analysis Details

For the analysis of the targeting capabilities in Figure 4, we ran two attacks:

1. MALT and APGD, with the same $a = 9$, $c = 100$ hyperparameters
2. APGD attack with naive targeting to top 9 targets, as it performed in the targeted version of APGD within AutoAttack.

We observe the number of successful attacks for each targeting method and for each ordered target suggested. For naive targeting columns, the targets are ordered by the corresponding class confidence level given by the model. For the MALT columns, the targets are ordered according to the MALT algorithm.

---

[6]https://github.com/RobustBench/robustbench
[7]https://github.com/fra31/auto-attack

