# OpenReview forum: "MALT Powers Up Adversarial Attacks"
_NeurIPS.cc/2024/Conference — NeurIPS 2024 poster_

### Official Review · Reviewer_hP7Q · 2024-07-03

**Soundness:** 4
**Presentation:** 3
**Contribution:** 3
**Rating:** 7
**Confidence:** 4

**Summary:**

AutoAttack is a highly successful image-based adversarial attack method that combines targeted and untargeted approaches to effectively target a wide range of models. For targeted attacks, AutoAttack selects 9 adversarial target classes based on the model's confidence levels. However, this restriction is imposed due to computational constraints. Notably, the authors of this paper argue that picking target classes based on the model's confidence levels is not an optimal approach. They demonstrate that for linear classifiers, given an input instance, the best target class minimizes the ratio of distance to the target class and the norm of the difference between the gradients. Additionally, they show that deep neural networks (DNNs) are almost linear in their mesoscopic region, indicating that this approach can also be applied to DNNs. Building on this insight, the authors propose a new method called MALT, which empirically outperforms AutoAttack and achieves faster adversarial perturbation discovery.

**Strengths:**

1.The study reveals a significant flaw in the AutoAttack method: the optimal target class for linear classifiers is not determined by the model's confidence levels. This fundamental finding highlights a limitation of AutoAttack and underscores its potential shortcomings. Specifically, AutoAttack may falter in certain scenarios due to its reliance on selecting the top 9 classes for targeted attacks, which can overlook the most suitable class if it does not fall within this range. By recognizing that linearity holds true in the mesoscopic region, the authors of the current paper are able to refine and improve upon AutoAttack. Their proposed approach, MALT, demonstrates empirical superiority over its predecessor while achieving faster adversarial perturbation discovery.
2.The authors also conduct a complexity analysis to compare the efficiency of MALT with AutoAttack. As MALT relies solely on APGD-based attacks, it exhibits a significant speed advantage over AutoAttack. Specifically, when applied to the Imagenet dataset, MALT demonstrates an average speedup of 5 times compared to AutoAttack, making it a more efficient and practical solution for adversarial attack detection.
3. The empirical findings presented in this paper are robust and convincing. The authors demonstrate that MALT consistently outperforms AutoAttack, with its success rate of finding adversarial examples never trailing behind and often surpassing AutoAttack. Moreover, MALT exhibits a significant speed advantage, allowing it to identify adversarial examples more efficiently. To validate their approach, the authors selected a value of c=100 for the Imagenet dataset, but their empirical analysis reveals that even higher values (e.g., c=1000) do not necessarily yield better results. This finding underscores the effectiveness of MALT's design. Furthermore, they empirically demonstrate that APGD is an excellent choice for their attack strategy.

**Weaknesses:**

1. While MALT's primary innovation is rooted in its refined class selection process, which sets it apart from AutoAttack, this refinement may somewhat temper the overall novelty of this manuscript.
2. The authors have chosen a specific value of c (c=100) and demonstrated empirically that this value is sufficient for the Imagenet dataset. However, the selection of an optimal value of c that generalizes well across various datasets remains unclear, leaving room for future exploration and refinement.
3. In line with their earlier findings, the authors have empirically demonstrated that APGD outperforms alternative strategies in the context of MALT. While this suggests a strong case for APGD as a suitable choice for MALT, it remains to be seen whether this advantage will hold across diverse settings and datasets.

**Questions:**

1. The authors showed that MALT leans towards the top-1. Should we select a=1 in all cases then?
2. Do the authors have any intuition behind why APGD is better than FAB for MALT?

---

> ### Author Rebuttal · Authors · 2024-08-06
>
> We thank the reviewer for the positive and thorough review.
>
>
> "**While MALT's primary innovation is rooted in its refined class selection process, which sets it apart from AutoAttack, this refinement may somewhat temper the overall novelty of this manuscript.**"
>
> We acknowledge that the main contribution of MALT is the introduction of a refined class selection process and its use with existing attacks such as APGD. We believe that although APGD is already widely used, the addition of MALT significantly improves it. We are sorry if we didn’t understand this point and would be happy to elaborate further.
>
>
> "**The authors have chosen a specific value of c (c=100) and demonstrated empirically that this value is sufficient for the Imagenet dataset. However, the selection of an optimal value of c that generalizes well across various datasets remains unclear, leaving room for future exploration and refinement.**"
>
> It is an interesting question to optimize the value of c, thus further reducing the running time of MALT. However, we believe it would be very beneficial if the value of c can be universal across all datasets and models, and thus be treated as a constant rather than a hyperparameter that needs tuning. In section 5.1, lines 314-325 we study the effect of the value of c, by choosing c=1000 (i.e. all the possible classes of Imagenet), and show that the robust accuracy doesn’t improve. This indicates that a value of c=100 is indeed a proper choice for all our experiments, at least as an upper bound, and we admittingly haven’t attempted to optimize it further.
>
> "**In line with their earlier findings, the authors have empirically demonstrated that APGD outperforms alternative strategies in the context of MALT. While this suggests a strong case for APGD as a suitable choice for MALT, it remains to be seen whether this advantage will hold across diverse settings and datasets.**"
>
> This is a good point. We have tested MALT combined with the targeted FAB attack (lines 297-299), and it performed worse than with APGD with the DLR loss. We additionally conducted another experiment for the rebuttal phase, using MALT with APGD and the CE loss (instead of DLR) on the SWIN-L model. The robust accuracy against MALT with APGD and CE loss is 60.52%, for Autoattack it is 59.9% and for MALT with APGD and DLR loss it is 59.84% (lower is better). We will add this experiment in the final version.
> It is indeed an interesting research direction to study other targeted attacks and experiment with MALT beyond the datasets that appear in the RobusBench benchmark.
>
> "**The authors showed that MALT leans towards the top-1. Should we select a=1 in all cases then?**"
>
> Although MALT does lean towards the top-1 better than naive top-k targeting, there are still images where the attacked target is not the top-1. We used top-9 according to the MALT class selection to align with the current targeting (top-9 according to the highest model confidence). Thus a=9 seems to be a good choice, although it could possibly be further optimized to further improve the running time of MALT.
>
> "**Do the authors have any intuition behind why APGD is better than FAB for MALT?**"
>
> We think that, in general, a targeted FAB attack is less successful than a targeted APGD attack. This can be seen in the experiments done in [Croce and Heine 2020] (Table 1). Hence, this difference between APGD and FAB is probably unrelated to MALT. We believe this is also true for the additional experiment done in the rebuttal phase with APGD and CE loss, which underperforms compared to APGD with the DLR loss.

---

> > ### Comment · Reviewer_hP7Q · 2024-08-11
> > **Response to author rebuttal**
> >
> > I thank the authors for going through my review and responding to my comments. However, after going through the other reviews and author rebuttals, I have decided not to update my score at this point.

---

### Official Review · Reviewer_fhJA · 2024-07-08

**Soundness:** 4
**Presentation:** 4
**Contribution:** 1
**Rating:** 3
**Confidence:** 3

**Summary:**

This paper introduces MALT, a heuristic technique for selecting target classes for adversarial perturbations. The intuition behind MALT is to order attack targets in the order of high row norm in the jacobian. They show that this can beat an AutoAttack baseline with much less compute for CIFAR and ImageNet classification tasks.

**Strengths:**

S1: The core result in tables 1 and 2 is compelling. The speedups are impressive.

S2: I liked the writing and clarity, but I have a few questions below.

**Weaknesses:**

W1: This paper seems to be behind its time. It only experiments with image classification including CIFAR. I don't really fault the paper for this, but modern image and other classification problems are just addressed with much bigger models. I don't see why it would be necessarily hard to implement this. If I ask myself "has this paper convincingly done experiments at a scale/setting that seem convincing that MALT would be useful for modern problems" I would say no.

W2: Section 4.1 seems pointless -- just proving something to prove something. I don't see any value in proving something here about a two-layer network. Conditional on section 5 already existing, I think that section 4 is of no interest.

**Questions:**

Q1: Why are these types of targeted attacks needed? In general, the purpose of a targeted attack is to make the model do a specific thing that the adversary wants. So why do targeted attacks if the attacker doesn't care about what the target it. Does it make the attack more efficient? Why not just do untargeted attacks? Why isn't this a baseline? I might be unfamiliar with the background lit, but has it been empirically, convincingly established that dynamically selecting target classes is more efficient than simply doing an untargeted attack in the first place?

Q2: What does "interval" mean in table 1?

Q3: In what sense is AutoAttack the SOTA? Says who? Why were no other baselines tested against?

**Limitations:**

L1: Model size & scale, problem realisticness, and demonstrated value in realistic applications. My main challenge with this paper is that I don't feel that there have bee many experiments to convincingly demonstrate value of MALT for realistic, useful applications in 2024.

---

> ### Author Rebuttal · Authors · 2024-08-06
>
> We thank the reviewer for the thorough review.
>
> "**W1: This paper seems to be behind its time.**"
>
> We first emphasize that all our experiments are done against the top robust models according to Robustbench, which is the de facto standard benchmark in this field. All the models we tested on are from 2024 or 2023 and are large-scale. For example, Swin-L [Liu et al. 2023] is the current best robust model for Imagenet, it is based on transformers and contains 187 million parameters. Our experiments are done for the Imagenet and CIFAR-100 datasets, we note that Robustbench includes only those datasets as standard benchmarks. It is a legitimate criticism that other datasets should be included as standard benchmarks in the adversarial attack research field, but this is relevant to all papers published in this field and not particular to ours.
>
>
> "**W2: Section 4.1 seems pointless**"
>
> Our method is based on an analysis of mesoscopic linear models, as explained in Section 3. We believe it is important and interesting to study whether this analysis is valid theoretically. Two-layer neural networks are already highly non-linear, thus studying the mesoscopic linearity properties in such networks is in itself challenging and interesting. A similar result was proven before only for networks with random weights [Bubek et al. 2021], and we extend it to trained networks under certain assumptions.
>
> We also believe that this study of mesoscopic linearity can be of independent interest beyond the scope of adversarial attacks, as it unveils a certain intriguing phenomenon about the optimization landscape of neural networks.
>
> "**Q1: Why are these types of targeted attacks needed?**"
>
> In earlier days of studying adversarial attacks, this was a very important question. Indeed, targeted attacks have been shown to be more effective than untargeted attacks, for a great study on this subject we refer to [Croce and Heine 2020]. Also, note that in AutoAttack , which is a composition of four different attacks, two are targeted and two are untargeted (lines 153-159). Since our attack performs better than Autoattack, in particular, it performs better than the current top untargeted attacks.
>
> We also emphasize that  MALT runs in an untargeted fashion, just like APGD-T with top-k classes runs in an untargeted fashion. Namely, given an image to attack, the attack itself does not target some predetermined class but rather automatically choses the classes that are easiest to attack according to the score given by MALT.
>
> "**Q2: What does "interval" mean in table 1?**"
>
> In Figure 1 we divide the adversarial perturbation into 100 intervals and present the change over time in the output’s confidence for each class. We will add this explanation in the final version.
>
> "**Q3: In what sense is AutoAttack the SOTA?**"
>
> AutoAttack is widely considered to be the state-of-the-art adversarial attack, which in itself is an ensemble of four different attacks. It is the standard benchmark to test against robust models in the official RobustBench benchmark.
>
> "**L1: Model size & scale**"
>
> Please see our answer to W1.

---

> ### Comment · Reviewer_fhJA · 2024-08-09
> **Thanks + reply**
>
> W1 -- I think that my main issue is with models, not datasets. I still wouldn't predict much practical value for the aforementioned reasons.
>
> W2 -- I still think that 4.1 contributes no marginal value. But I see where you're coming from. Honestly, my issue here isn't really with this paper so much as the field in general.
>
> Qs -- thanks
>
> Overall, I think I would stick at a 3. I understand that I'm the cranky reviewer this time, and I respect the other reviews, but I think it's important to emphasize that I just don't see the practical value of this work in 2024. Good luck. I'm open to replies.

---

> > ### Author Response · Authors · 2024-08-09
> >
> > We appreciate the reviewer’s feedback.
> >
> > Modern papers that study adversarial attacks are expected to test their attack against the top robust models from Robustbench, and against other successful attacks, which we did. Attacking any other models, not specially trained to defend against those attacks, will not be a “fair” comparison.
> >
> > Beyond the scope of our paper, adversarial attacks are a very important research topic in practice these days. Developing robust models and new attack methods is very practical in the industry (in 2024), beyond only academic interest. For example, attacks such as AutoAttack are widely used in many scenarios in the industry, e.g., companies demonstrating robust predictions for their customers, testing models for robustness internally, or for robust training. This is due to the high demand for robust ML and the simple open-source implementation of adversarial attacks, MALT included.
> >
> > We are uncertain about how to further enhance the relevance in your opinion of our work for 2024.
> >
> > Could you please provide more specific guidance on how we might improve the paper to increase its practical value for 2024? Understanding your perspective would greatly assist us in refining our work.

---

> > > ### Comment · Reviewer_fhJA · 2024-08-13
> > > **clarification**
> > >
> > > I work on attacks too -- I think it's valuable. My reservation involves the scale of the experiments and the lack of connection to more competitively accomplishing practical tasks of interest in models that are used in real applications.
> > >
> > > Good luck to the authors. Congrats on an interesting paper. Sorry for being the cranky reviewer, but I enjoyed reviewing still and I hope you all keep up the good research.

---

### Official Review · Reviewer_BXuQ · 2024-07-12

**Soundness:** 3
**Presentation:** 3
**Contribution:** 3
**Rating:** 6
**Confidence:** 3

**Summary:**

The paper presents a novel adversarial targeting method, Mesoscopic Almost Linearity Targeting(MALT), based on local almost linearity assumptions. The proposed attack wins over the current state of the art AutoAttack on the standard benchmark datasets CIFAR-100 and Imagenet and for different robust models. The proposed attack uses a five times faster attack strategy than AutoAttack's while successfully matching AutoAttack's successes and attacking additional samples that were previously out of reach. The paper proves formally and demonstrates empirically that the proposed targeting method can apply to non-linear models.

**Strengths:**

The paper has good originality, quality, clarity, and of important significance.

**Weaknesses:**

The proposed MALT's performance advantage over AutoAttack is not obvious.

**Questions:**

1.Does the proposed method be effective for untargeted method?
2.The proposed MALT's performance advantage over AutoAttack is not obvious as shown in Table 1 and Table 2.
3.Deepfool is also the minimal distance attack method, what's the difference between Deepfool and the proposed MALT?
4.small typo errors,line 30,Provides.

**Limitations:**

The experimental performance of the proposed method is not that good.

---

> ### Author Rebuttal · Authors · 2024-08-06
>
> We thank the reviewer for the thorough review.
>
> "**The proposed MALT's performance advantage over AutoAttack is not obvious.**"
>
> We acknowledge that the additional attacked images are not numerous, resulting in a marginal improvement in terms of successful attacks over Autoattack. However, this is a very different concept from an improvement in other metrics in machine learning where indeed marginal improvement may not be significant. The reason is that for every image that AutoAttack successfully attacks, MALTs also successfully attack, while also attacking additional images that are out of reach for AutoAttack. Hence, there is a strict inclusion in the successful attacks. This is while improving running time by a factor of 5 on the entire imagenet attack test set. Also, this improvement is consistent across all the 9 tested models, and two benchmark datasets.
>
> Also, note that we haven’t altered any other properties or any hyperparameters other than improved targeting. Compared to AutoAttack, we used just one out of the four attacks included in its ensemble (namely, APGD-T), and changed its targeting method. Therefore, the time gain and the attack improvements are all due to MALT.
>
> "**Does the proposed method be effective for untargeted method?**"
>
> Yes, actually MALT is a method to choose the top-k targets. Thus, by choosing k targets MALT runs as an untargeted method, just like APGD with top-k classes runs in an untargeted fashion, meaning that choosing the targets is not part of the attacker's input but rather a part of the algorithm. We note that APGD can run completely untargeted, i.e. doing gradient steps that are not targeting any target class. MALT is a method to improve the choice of easier targets to attack, thus it is inherently different from APGD in this sense. Also, targeted attacks have been shown to perform better than untargeted attacks (see for example [Croce and Heine 2020]), thus we focus in this paper on the targeted version of APGD.
>
>
> "**The proposed MALT's performance advantage over AutoAttack is not obvious as shown in Table 1 and Table 2**"
>
> Please see our answer to the first comment.
>
> "**Deepfool is also the minimal distance attack method**"
>
> Deepfool can be compared to APGD as an attack that performs gradient steps. However, MALT is a method to improve the targeting in targeted attacks. Thus, MALT may also improve the performance of Deepfool. We focused on APGD since this is a widely used standard attack, which is part of AutoAttack, the current state-of-the-art adversarial attack.
>
> We will fix the typos in the final version

---

> > ### Comment · Reviewer_BXuQ · 2024-08-11
> >
> > Thanks to the authors for their reply, I have no more questions.

---

### Official Review · Reviewer_WTPi · 2024-07-12

**Soundness:** 3
**Presentation:** 3
**Contribution:** 3
**Rating:** 6
**Confidence:** 3

**Summary:**

Several evasion attacks seek untargeted adversarial examples by performing multiple runs with a targeted loss against different target classes. This process often leads to better results with respect to using untargeted losses, as the optimization process might be more stable. Until now, the choice of the considered target classes has been made by simply picking the top-k output scores - excluding the true class. This paper presents a novel method to perform this choice based on a score computed by normalizing the difference between the candidate target and the true class scores by the norm of the input gradients with respect to them. Using this score instead of the common naive approach on APGD-T is shown to improve the attack performance while reducing its computational cost.

**Strengths:**

Although based on existing previous works, this work presents a novel idea that addresses an underexplored aspect of evasion attacks to the best of my knowledge. The problem of improving the performance of these attacks is well-known and very relevant, as - apart from very few settings where certified defenses can be applied - the robustness of machine learning algorithms can only be evaluated with empirical methods, and there are no formal guarantees on the provided results which might overestimate it. Thus, stronger attacks help a reliable robustness assessment. The proposed approach is well presented and formulated (including theoretical and empirical justifications), reports promising results, and can be applied to any attack.

**Weaknesses:**

Some aspects of the attack evaluation should be clarified or integrated with additional assessments. As the improvements provided by MALT with respect to AutoAttack are quite marginal, the experiments should clearly show that these improvements are exclusively due to the applied method.

Even if comparing MALT with the entire AutoAttack suite is very important, it is interesting here to provide insights on how much MALT improves the standard APGD-T algorithm. In addition, the authors could perform some additional tests to validate their results further (see questions). I don't expect that all the required experiments will be performed, but I will suggest them as I think they could strengthen this work.

The authors claim multiple times that their method is five times faster than AutoAttack. This value is estimated by comparing the total number of forward and backward passes required by MALT and AutoAttack and then empirically computed by measuring the runtime. However, the empirical measurements are unreliable as they are highly influenced by several factors. Thus, I think it would be sufficient to consider the number of forward and backward passes (separately). Regarding the provided values in Sect. 3.3, the authors report the number of forward and backward passes for a worst-case scenario where all the attacks of the AutoAttack suite are executed. This should be clearly stated. Additionally, the experiments should report the actual number of forward/backward passes of AutoAttack, as it is likely that not all the attacks are executed for many samples.

**Questions:**

- Can you compare the MALT performance to APGD-T with random restarts and APGD-T with 9 randomly chosen target classes?
- How does MALT perform using APGD with targeted CE loss? (If you perform these experiments, it would also be interesting to see APGD with random restarts and randomly chosen target classes.)
- These two papers also provide improvements with respect to AutoAttack (in both performance and efficiency). Can you compare MALT with them?


[a] Liu, Y., Cheng, Y., Gao, L., Liu, X., Zhang, Q., & Song, J. (2022). Practical Evaluation of Adversarial Robustness via Adaptive Auto Attack. 2022 IEEE/CVF Conference on Computer Vision and Pattern Recognition (CVPR), 15084-15093.

[b] Yao, C., Bielik, P., Tsankov, P., & Vechev, M.T. (2021). Automated Discovery of Adaptive Attacks on Adversarial Defenses. Neural Information Processing Systems.

**Limitations:**

Limitations are discussed.

---

> ### Author Rebuttal · Authors · 2024-08-06
>
> We thank the reviewer for the thorough review.
>
> "**Some aspects of the attack evaluation should be clarified or integrated with additional assessments. As the improvements provided by MALT with respect to AutoAttack are quite marginal, the experiments should clearly show that these improvements are exclusively due to the applied method.**"
>
> We emphasize that we haven’t altered any properties or any hyperparameters of APGD-T other than improved targeting. Compared to AutoAttack, we used just one out of the four attacks included in its ensemble (namely, APGD-T), and changed only its targeting method. Therefore, the time gain and the attack improvements are all due to MALT.
>
> We acknowledge that the additional attacked images are not numerous, resulting in a marginal improvement in terms of successful attacks over AutoAttack. First, we note that the dramatic running time improvement, while also improving the attack success rate, is in itself a notable contribution. In addition, this attack success rate improvement is due to an inherent weakness of the existing widely used targeting methods and is neither dataset\model-specific nor depends on some fine-tuned hyperparameters. Using nothing but better targeting, not only do we gain the improvement over APGD-T gained by AutoAttack, but also add additional improvement by attacking images that were previously out of reach for both naively targeted APGD-T and AutoAttack.
>
> "**Even if comparing MALT with the entire AutoAttack suite is very important, it is interesting here to provide insights on how much MALT improves the standard APGD-T algorithm.**"
>
> We made this comparison as the reviewer suggested. This experiment is done on the Swin-L model [Liu et al. 2023] and Imagenet dataset, this is the top robust model for Imagenet according to Robustbench. APGD-T with standard targeting achieves 59.94% robust accuracy, Autoattack achieves 59.9% robust accuracy and APGD-T with MALT targeting achieves 59.84% robust accuracy (lower is better). There is a strict inclusion between the attacks, namely every image that APGD-T with standard targeting successfully attacks, Autoattack also attacks, and similar for Autoattack and APGD-T with MALT. We acknowledge that at today’s level of adversarial attack, it is very difficult to significantly improve over current attacks. However, we believe the main advantage of our method is to outperform the current state-of-the-art on every tested model, while also significantly improving its running time. If the reviewer thinks this is an important comparison, we can add it also to the other models in our paper in the final version.
>
>
> "**The authors claim multiple times that their method is five times faster than AutoAttack. This value is estimated by comparing the total number of forward and backward passes required by MALT and AutoAttack and then empirically computed by measuring the runtime. However, the empirical measurements are unreliable as they are highly influenced by several factors.**"
>
> For the empirical measurements of the running time of AutoAttack and MALT, we used 5 different batches while performing the experiments on the exact same GPU. All the details are in lines 288-295. We agree that this empirical measurement might not be precisely exact, but since it is consistent across 9 different models, and for five different batches for each model (while having relatively small variance) we believe this is a reasonable conclusion. We acknowledge that the analysis in Section 3.3 is for a worst-case scenario, and we’ll emphasize it in the final version.
>
> "**Can you compare the MALT performance to APGD-T with random restarts and APGD-T with 9 randomly chosen target classes?
> How does MALT perform using APGD with targeted CE loss? (If you perform these experiments, it would also be interesting to see APGD with random restarts and randomly chosen target classes.)**"
>
> We thank the reviewer for these helpful questions. We performed these experiments for the rebuttal as the reviewer suggested, and will add them in the final version. Here are the details:
>
> (1) We performed an APGD-T attack with 9 random targets on the Swin-L model [Liu et al. 2023] (which is the top robust model for Imagenet according to Robustbench). This attack was significantly less successful, achieving a robust accuracy of 78.54%, compared to 59.9% with Autoattack and 59.84% with MALT and APGD-T with a DLR loss (lower is better).
>
> (2) We performed an APGD-T attack with CE loss combined with the targeting of MALT. Again, the attacked model is Swin-L, and the dataset is Imagenet. This resulted in a robust accuracy of 60.52%, which is close to APGD-T with the DLR loss and MALT (59.84%), but still underperforms.
>
> (3) Regarding random restarts, all our experiments that include APGD have a random restart which is the default setting of APGD. According to [Croce and Heine, 2020], which introduced the APGD attack, additional random restarts do not improve the success rate.
>
> "**These two papers also provide improvements with respect to AutoAttack (in both performance and efficiency). Can you compare MALT with them?**"
>
> Thank you for the suggestion. Due to time constraints in the rebuttal phase, we did not run those experiments, but we will consider adding these comparisons in the final version. Both papers include also targeted attacks, thus MALT can also be used as an off-the-shelf targeting method to improve these methods. We also note that both papers consider only the CIFAR dataset and not Imagenet. The reason may be because these are adaptive methods, which adapt to the model and dataset and require a very long precalculation to allow this adaptivity. These precalculations may be too long to run for Imagenet and for the current state-of-the-art robust models which are significantly larger than the models that are used in those papers.

---

> > ### Comment · Reviewer_WTPi · 2024-08-09
> > **Response to rebuttal**
> >
> > I thank the authors for considering my suggestions and performing additional experiments, which, in my opinion, strengthened the contribution of this paper. Although I still believe that it would have been better to report the actual number of forward/backward passes of AutoAttack instead of the runtime, I understand that this would have required running all the experiments again. I will consider raising my score anyway.

---

### Decision · Program_Chairs · 2024-09-25

**Decision:**

Accept (poster)

**Comment:**

This paper proposes a new method for targeted adversarial attacks. The method is shown to be (marginally) more effective than AutoAttack while being significantly faster. Overall, reviewers appreciated the novelty of the idea as well as the intuitive and theoretical justification, as well as the robustness of the empirical evaluation. One reviewer was concerned about the size and scale of the experiments, but given that no concrete suggestion was given to the authors I am inclined to accept the paper. I encourage the authors to address any and all reviewer comments and suggestions, specifically about more accurately reporting speedups, methods for setting hyperparameters, and additional experimental comparisons (where applicable).